



# Parameter-state ensemble data assimilation using Approximate Bayesian Computing for short-term hydrological prediction

Bruce Davison[1], Vincent Fortin[2], Alain Pietroniro[1], Man K. Yau[3], and Robert Leconte[4]

[1]Environment and Climate Change Canada, Saskatoon, Saskatchewan, Canada.
[2]Environment and Climate Change Canada, Montreal, Quebec, Canada.
[3]McGill University, Montreal, Quebec, Canada.
[4]Université de Sherbrooke, Sherbrooke, Quebec, Canada.

*Correspondence to:* Bruce Davison (bruce.davison@canada.ca)

**Abstract.** The main sources of uncertainty in hydrological modelling can be summarized as structural errors, parameter errors, and data errors. Operational modellers are generally more concerned with predictive ability than model errors, and Data Assimilation (DA) methods are commonly employed to merge models with observations to improve predictive ability. This paper presents an example of Approximate Bayesian Computing (ABC), or a simplified Particle Filter (PF), to simultaneously

assimilate model states and parameters, calling the method Parameter-State Ensemble DA (P-SEDA). The case study is from June to October, 2014 for a small (1 324 km$^2$) watershed just north of Lake Superior in Ontario, Canada using the Canadian semi-distributed hydrologic land-surface scheme MESH. The study examines how well the approach works given various levels of certainty in the data; beginning with certainty in the streamflow and precipitation, followed by uncertainty in the streamflow and certainty in the precipitation, and finally uncertainty in both the streamflow and precipitation. The approach is

found to work in this case when streamflow and precipitation is fairly certain, while being more challenging to implement in a forecasting scenario where future streamflow and precipitation is much less certain. The main challenge is determined to be related to parametric uncertainty and ideas for overcoming this challenge are discussed.

## 1 Introduction

A fundamental problem making good streamflow predictions in process-based models rests with the various sources of un-
certainty in modelling the flow. These sources of uncertainty have been described in a number of papers (e.g. Beck, 1987; Krzysztofowicz, 2001; Vrugt et al., 2005; Liu and Gupta, 2007; Velázquez et al., 2009). In particular, Liu and Gupta (2007) consider a general framework of seven model components which include the system boundary ($B$), inputs ($u$), initial states ($x_0$), parameters ($\Theta$), structure ($M$), states ($x$), and outputs ($y$). Five of these model components ($B$, $u$, $x_0$, $\Theta$, and $M$) must be predefined and their uncertainties cascade to $x$ and $y$. Since the inputs, initial states, and observations used to verify the model

outputs can often be considered as data errors, and the system boundary can be considered a source of structural uncertainty, the main sources of errors in hydrologic modelling can be summarized as structural errors, parameter errors, and data errors. Operational hydrological models are generally more concerned with predictive ability than correctness of the model structure (Gupta et al., 2008, p 3804). As such, parameter and data errors are often the focus for operational hydrological predictions.





Within this context of managing parameter and data uncertainty, it is the purpose of this paper to propose a novel approach to short-term hydrological prediction in a relatively small, data-sparse watershed ($1{,}324$ km$^2$). The approach involves using a hydrologic land-surface-scheme (H-LSS) and simultaneous estimation of parameters and state variables through Data Assimilation (DA) using Approximate Bayesian Computation (ABC, Biau et al., 2015), or a simplified version of the Particle Filter (PF, Arulampalam et al., 2002).

DA is one way to improve hydrological predictions by merging models with observations, and can include methods that help resolve problems related to estimating states, assessing parameters and identifying the appropriate model structure (Liu and Gupta, 2007). Most applications of DA focus on merging state variables in a model with corresponding observations, while a few methods combine state and parameter estimation to improve predictions (e.g. Vrugt et al., 2005; Moradkhani et al., 2005a, b; Drécourt et al., 2006; Labarre et al., 2006; Qin et al., 2009; Nie et al., 2011; Xie and Zhang, 2013; Bi et al., 2014).

The strategies for combined state and parameter assimilation generally fall into three main categories (Liu and Gupta, 2007). One strategy, such as that used by Vrugt et al. (2005), is to use standard techniques to simultaneously optimize parameters and assimilate states. In this strategy, an outer loop is used to optimize parameter sets while an inner loop is used to assimilate the state variables for each calibration parameter set at each time-step. Another strategy is to use dual filters (e.g. the dual Ensemble Kalman Filter or dual Particle Filter) to update states and parameters independently (e.g. Moradkhani et al., 2005a, b; Qin et al., 2009). In these cases, the parameters and states are continuously updated as new observations become available. The third strategy is most often called "state augmentation" and uses regular data assimilation methods where parameters are considered state variables and added to the state vector (e.g. Drécourt et al., 2006; Nie et al., 2011; Bi et al., 2014). A single filter is then used to update the parameters and states simultaneously. One drawback of traditional DA (of states only) and of the aforementioned parameter and state DA methods, however, is that the resulting parameters and states are not necessarily compatible with one-another.

In this paper, a new and very simple method of simultaneous state and parameter DA, that ensures parameter and state compatibility, is presented for short-term hydrological ensemble prediction (up to 3 days). We call this DA-approach the Parameter-State Ensemble Data Assimilation (P-SEDA) filter and make use of ABC, which is also a simplified PF. The approach is described with the intention of making clear how to implement the filter with a wide variety of models in data-rich or data-sparse watersheds, and examined here using a parameter-intensive hydrologic land-surface scheme in a data sparse watershed. The case studies include model structural and parameter errors, which are inevitable regardless of the model being used or the basin being modelled, to evaluate the ability of the filter to work under such conditions.

The intial case studies are hindcasting exercises that reduce data and model structural uncertainty as much as possible, followed by a more realistic forecasting example (albeit in hindcasting mode) that incorporates data input uncertainty using Environment and Climate Change Canada's (ECCC's) meteorological Regional Ensemble Prediction System (REPS) to drive the model.



## 2   Methodology

### 2.1   The Parameter-State Ensemble Data Assimilation (P-SEDA) Filter

The P-SEDA filter works in the following manner. First, a number of parameter sets ($M$) are pre-defined to be used for continuous simulation with a model. Filtering criteria are used to determine which of the parameter sets and their associated

state variables will be used to generate an ensemble of streamflows for analysis in a projection period. The analysis is completed and the process repeated for the next appropriate time-step in the continuous simulations. In this manner, both the parameters and states are drawn from the entire $M$ simulations for the projection period. This is very similar to traditional particle filtering methods.

A traditional particle filter has the following four steps: 1) generate an initial set of particles (or parameter sets) and run the

model for a short time (e.g. one timestep) to produce model output for the variable of interest, 2) assign a weight between zero and one to each particle such that higher weights are given to parameter sets that produce model outputs more closely matching the observed variable, 3) resample the parameter space with respect to the weights (i.e. produce a new set of particles with parameters that are closer to the parameter sets that produced higher weights), and 4) propogate the new particles using the model, thus repeating the cycle. The approach presented here is the same, but without resampling and always returning to the

original particles as updated by the model and assigning a weight of zero or one to each particle based on the filter (i.e. using a rectangular filter).

For a single filter-projection period, this approach is also described by ABC. As described by Vrugt and Sadegh (2013), for scenarios focused on parameter uncertainty (as is the case in this paper) the posterior parameter distribution $p(\theta|y)$ given the streamflow $y$ is estimated using Bayes theorem:

$$p(\theta|y) = \frac{p(\theta)p(y|\theta)}{p(y)}$$

where $p(\theta)$ is the prior distribution of parameters, $p(y)$ is the normalization constant, and $p(y|\theta) \equiv L(\theta|y)$ is the likelihood function. In situations where the likelihood function cannot be computed (again, as is the case in this paper) the likelihood is approximated using a model.

Biau et al. (2015, Algorithm 2) provides a widely-used algorithm for the approach, shown below as Algorithm 1. The

symbols not yet described in Algorithm 1 include a statistic representing the observations, $s_0$, and the statistic representing the simulations $s(y_i)$.

In the context of the P-SEDA filter for hydrological prediction, $\theta_i$ is the $i^{th}$ parameter set. The prior $p(\theta)$ represents the infinite possible parameter sets based on ranges selected by the user and the likelihood is approximated by the model used to generate the $y_i$ simulated streamflow values (giving rise to the "Approximate" in ABC). In Biau et al. (2015), only the $k_M$-

nearest neighbors between the statistic representing the observations, $s_0$, and simulations, $s(y_i)$, for the filter period are kept for analysis. The case study presented below alters the selection criteria slightly by using a distance function (Root Mean Squared




---

**Algorithm 1** Pseudo-code of a generic ABC algorithm

---

**Require:** A positive integer $M$ and an integer $k_M$ between 1 and $M$.

**for** i = 1 to $M$ **do**

generate $\theta_i$ from the prior $p(\theta)$;

generate $y_i$ from the approximate likelihood $p(y|\theta)$

**end for**

**return** The $\theta_i$'s such that $s(y_i)$ is among the $k_M$-nearest neighbors of $s_0$.

---

Error) to determine the discrepancy between the observations and the simulations, rather than independent statistical properties (such as mean and standard deviation) of the two.

In the P-SEDA approach, we can disregard the notion of finding a distribution of parameter sets that fits the entire streamflow record of interest. Instead, we look for a set of plausible parameter sets, locate a certain number of these that generate the "best" results for the filter period under consideration, and then evaluate how well these parameters and states perform for the projection period. The process is then repeated to find new parameter sets and states for consideration in successive projection periods. There are a number of ways in which filter and projection periods can be formulated. Four such formulations are described in section 2.7, which should clarify the generic process described in this paragraph. As already mentioned, this approach can be considered as a simplified PF or ABC.

## 2.2 Case Study Basin Description

The study watershed is 1,324 km$^2$ and is drained by the Little Pic River near Coldwell, Ontario, Canada, just north of Lake Superior. The streamflow gauge (02BA003) is between the communities of Terrace Bay and Marathon and has been operated by the Water Survey of Canada from 1972 to the present. There are no precipitation measurements in the basin, but the surrounding region's annual precipitation ranges from 654 to 879 mm, with the mean summer rainfall ranging from 231 to 298 mm (Crins et al., 2009). The mean annual temperature ranges from $-1.7$ to 2.1 °C (Mackey et al., 1996). The sub-surface sits on Precambrian Shield with significant amounts of volcanic rock, greenstone, siltstone and shale (Sutcliffe, 1991). The dominant landcover in the basin is mixed forest, followed by coniferous forest, water, sparse forest and deciduous forest (Crins et al., 2009). The streamflow regime is characterized by frozen conditions through the winter months (November to April), but has been known to produce a spring freshet as early as March. Summer and autumn peaks can be on the same order of magnitude as the spring freshet, but are more often smaller. The peak flow is usually in May and the highest daily discharge recorded is 269 m$^3$/s on June 30, 2008.

## 2.3 The Semi-Distributed Hydrologic Land-Surface Scheme

The model used to simulate the streamflow is the semi-distributed hydrological land-surface scheme MESH (Pietroniro et al., 2007), configured with the Canadian Land-Surface Scheme (CLASS, Verseghy, 1991; Verseghy et al., 1993), the hydrologic



routing from WATFLOOD (Kouwen et al., 2002), and additional hydrological processes to better simulate surface and sub-surface lateral flow across the landscape to the river (Soulis et al., 2000, 2011).

The basin geophysical characteristics needed for MESH include a digital elevation model (DEM), landcover classification, and soil information. The DEM comes from the Canadian Digital Elevation Data (CDED) at a scale of 1:50,000 and based

on the NAD83 horizontal reference datum (Natural Resources Canada, 2015). The landcover classification comes from the LCC2000-V product originating from classified Landsat 5 and Landsat 7 satellite images and the soils information comes from the ecodistricts classification of the national ecological framework for Canada (Agriculture and Agri-Food Canada, 2015). The basin fits within ecodistrict 389 - Long Lake.

Table 1 shows the estimated percentages of each landcover present in the basin as defined by the LCC2000-V product.

Based on this classification, the two dominant landcovers are coniferous and broadleaf forest, which are often mixed. Without knowing more specific information about the landcover, the mixed forests are assumed to be fifty percent coniferous and fifty percent broadleaf, resulting in an estimate of forty-eight percent coniferous and thirty-nine percent broadleaf. These values are then arbitrarily rounded up to fifty percent coniferous and forty percent broadleaf in the model representation of landcover. The remaining ten percent of landcover inevitably includes parametric uncertainty due to the model's inability to properly represent

the eight percent of the basin that is covered by small lakes.

Figure 1 illustrates a) the location of the basin, b) ecodistrict boundary and model grid, c) river network and gauge location, and d) landcover. Sub-grid variability of each grid is handled via the CLASS tile with each grid being represented by a single ecodistrict GRU.

## 2.4 Forcing Data

The meteorological inputs for MESH include incoming shortwave radiation, incoming longwave radiation, precipitation, temperature, barometric pressure, specific humidity and wind speed. The timestep of the model is set to 30 minutes. For the first two case studies minimizing input data uncertainty, most of these meteorological inputs were derived from ECCC's Global Environmental Multi-scale (GEM) Numerical Weather Prediction (NWP) model (Côté et al., 1998a), stitching together the 6–17 hour UTC forecasts from twice-daily runs beginning in January, 2002 and linearly interpolating between hours to obtain half-

hourly values. Precipitation is obtained from the Canadian Precipitation Analysis (CaPA, Mahfouf et al., 2007; Lespinas et al., 2015), which is an assimilation of ground-based observations and GEM precipitation forecasts. For the third case study including forcing input data uncertainty, an ensemble of meteorological inputs is obtained from ECCC's Meteorological Regional Ensemble Prediction System (REPS). The REPS provides 72 hour forecasts twice daily.

## 2.5 Parameter Selection

H-LSS's contain many parameters and there is a large body of scientific literature examining various techniques for effectively estimating parameters (for a brief review, see Matott et al., 2009). The method that was used in this study is Latin Hypercube Sampling (LHS, McKay et al., 1979). Twenty-eight parameters were perturbed based on the results of a simple study (not





shown) comparing the perturbation of 6, 15 and 28 parameters. Table 2 shows the parameter values that were fixed during the simulations while Table 3 shows the ranges for parameters that were perturbed.

It is worth noting up-front that this approach to parameter perturbation is very inefficient. Sampling via LHS is a variation of uniform random sampling that is traditionally used in the generalized likelihood uncertainty estimation (GLUE) methodology
(Beven and Binley, 1992). Tolson and Shoemaker (2008) provide a very thorough explanation of the limitations of LHS and other methods of combatting the inefficiency of the tradtional GLUE uniform random sampling. The purpose of this study, however, is to examine the P-SEDA methodology. Implications of the parameter sampling methodology are examined in the discussion after the results are presented.

## 2.6 Projection Periods for Short-term Hydrological Prediction

This paper is focused on short-term hydrological ensemble prediction (up to 3 days), with an interest in using the ECCC meteorological REPS to force a more comprehensive H-EPS. As such, projection periods are defined as the three-day windows of time from the beginning of each ECCC-REPS run at 0 UTC and 12 UTC. The red and pink bars in Figure 2 illustrate twelve projection periods beginning on 0 UTC, July 20 to 12 UTC, July 25, 2014. The remainder of Figure 2 is described in section 2.7.3.

## 15 2.7 Ensemble Selection Methodologies

The total population of model runs is generated by setting $M$ to 10,000 in Algorithm 1 and using LHS to generate the 10,000 parameter sets from a uniform prior distribution of 28 parameters based on the ranges shown in Table 3. To approximate the likelihood, MESH is run with each of the 10,000 parameter sets in a continuous simulation mode to generate streamflow values ($y$) for the period of June 2002 to November 2014. The Data Assimilation is performed by filtering the total population
of 10,000 model runs to generate an ensemble of the 10 "best" model runs for each projection period. The following four ensemble selection methodologies, or ensemble data assimilation filters, are examined in this study and will be described shortly:

1. Minimized uncertainty filter

2. Bulk calibration filter

3. Preceding streamflow filter

4. Parameter and preceding streamflow filter

An initial evaluation of the P-SEDA filters requires some sources of uncertainty to be minimized. In particular, streamflow observations and precipitation uncertainty are considered; and questions around the model's ability to manage snow processes are simply avoided.

The quality of a streamflow observation is commonly known to be affected by ice, the occurrence of which is noted in the Water Survey of Canada database of streamflow. For the Little Pic River watershed, ice on the river can occur as early



as November and as late as April. In addition, prior to the spring freshet, the streamflow contains almost no information that could assist in predicting future streamflow. The future flow primarily depends on other land-surface characteristics such as snow water equivalent and frozen ground. Because the immediate interest is in evaluating the P-SEDA filters when snow or ice on the ground is not present, the analysis is only performed from June 1 to October 31 for selected years, with some qualitative

analysis beginning on May 1, 2014.

Each of the four filter configurations is described below.

### 2.7.1   Minimized Uncertainty Filter

Since there are no precipitation observations in the basin, but there are some precipitation gauges nearby which are used in the generation of CaPA, the filters are initially tested with reduced precipitation uncertainty by forcing the model with CaPA.

In addition, streamflow uncertainty is minimized by filtering with known streamflow for the time periods of interest. In other words, in this configuration, the filter period corresponds with the projection period and we use the observed streamflow to filter the parameter-state combinations to generate the ensemble. The resulting minimized uncertainty filter is a hind-casting excercise to determine if the parameter selection methodology (LHS) allows the model to produce streamflow values that match observations.

### 2.7.2   Bulk Calibration Filter

The bulk calibration filter represents a more traditional calibration and validation exercise in which the calibration period is the filter period and the validation period is the projection period. In our example, the top 10 daily RMSE values from the 10,000 LHS simulations for Jun 1, 2002 to June 1, 2009 are filtered to see how they perform for the projection periods from June 1 to October 31, 2014.

### 2.7.3   Preceding Streamflow Filter

Figure 2 illustrates the third filter considered. In this Figure, twelve filter periods are shown in orange and green for July 17 to July 25, 2014. The first filter period is represented by the orange bar near the top of the Figure beside the ABC1 label. This filter period runs from 0 UTC on July 17 to 0 UTC on July 20, 2014. During this three-day period, the "best" parameter sets are selected based on how the model simulates the observed streamflow. The simulations for these top performing parameter

sets are then extended for three more days, which is considered to be the projection period.

This process is then repeated 12 hours later. The second filter period is represented by the orange bar illustrated just below the first filter period, and labeled ABC2 in the Figure. This filter period runs from 12 UTC on July 17 to 12 UTC on July 20, 2014. During this three-day period, the "best" parameter sets are selected based on how the model simulates the observed streamflow. Although there is considerable overlap between the first and second filter periods, the second filter period begins

and ends 12 hours after the first filter period, producing a new ensemble of "best" parameter sets. The simulations for these



new parameter sets are then extended for three days, which is considered to be a new projection period that also begins and ends 12 hours after the previous projection period.

The process is then continually repeated every 12 hours as shown by the remainder of the bars shown in the rows labeled ABC3 to ABC12. Each instance of the filter and projection periods represents a single application of ABC, which is why each

row is labeled as such. We call this filter the preceding streamflow filter because the projection periods shown in red and pink occur immediately after the filter periods.

One important consideration, that becomes relevant in the analysis, is the six hours of precipitation that occurred on July 22 from 14 UTC to 20 UTC. This is illustrated by the small blue bar at the top and near the middle of Figure 2. Some of the projection periods "see" this precipitation event (illustrated by the green bars) and some do not (illustrated by the orange bars).

As will be explained more fully in section 2.9.2, the analysis is split according to the sub-periods of the filter and projection periods that a) occur during and just after the precipitation event (the light green and pink bars); and b) that occur when it is otherwise rain-free (orange, dark green and red bars).

### 2.7.4 Parameter and Preceding Streamflow Filter

The fourth filter considered is very similar to the preceding streamflow filter. However, to constrain the parameter space further

than determined by the LHS, the population of parameter sets is reduced by selecting a sub-set from the initial population of 10,000 parameter sets. The sub-set is selected by confining the parameter values based on model simulations that respond well during precipitation events in 2014. Details are provided in section 3.4 of the results.

### 2.8 H-EPS

Of course it is not often known for sure if precipitation will occur in the future, and certainly not the amount of precipitation

that will occur. As a result, ECCC's Meteorological REPS is also used with the June 1 to October 31, 2014 data in a hindcasting mode to examine how the PSEDA approach can be used in a forecasting context. At 00 UTC and 12 UTC every 12 hours from May 1 to October 31, 2014, the top 10 parameter-state pairs from the filter period and different projection periods are run for 3-days using the forcing data from the 20 members of the REPS, for a total of 200 H-EPS members. This analysis is performed for all filters except for the bulk calibration filter.

### 2.9 Verification

Demargne et al. (2010) differentiates diagnostic verification for evaluating the performance of a system from real-time verification for helping end-users make decisions about the future. The verification performed here is done for the first of these objectives; evaluating the performance of a system.



### 2.9.1 Verification of the Ensemble Selection Methodologies

First, a qualitative analysis is undertaken to take advantage of the human brain's ability to synthesize information. The results are then quantitatively verified using the Ensemble Verification System (EVS, Brown et al., 2010). To examine the quality of the ensemble mean when compared with the corresponding observation, the mean error (ME) is calculated. Then the quality of

5 the ensemble distribution is calculated using rank histograms. Finally, the skill relative to using the current streamflow as the forecast is calculated using the mean Continuous Ranked Probability Skill Score ($\overline{\text{CRPSS}}$).

The ME measures the average difference between a set of forecasts and corresponding observations. In this case, it measures the average difference between the mean average of the ensemble forecast ($\overline{Y}$) and the observation ($x$) as follows:

$$\text{ME} = \frac{1}{n} \sum_{i=1}^{n} (\overline{Y}_i - x_i)$$

The ME may be positive, zero, or negative. A positive value represents an ensemble mean that is positively biased while a negative error represents an ensemble mean that is negatively biased. A value of zero represents an absence of bias in the ensemble mean.

The rank histogram measures the reliability of an ensemble forecasting system. It involves counting the fraction of observations that fall between any two ranked ensemble members in the forecast distribution. For an ensemble forecast containing

15 $m$ ensemble members ranked in ascending order, there are $m-1$ spaces between any two ranked ensemble members and two spaces at the ends (above and below the ensemble forecast range) for a total of $m+1$ spaces ($s_1, \dots, s_{m+1}$). The corresponding observation $h$ for each ensemble forecast will fall within one of the spaces.

$$h_i = \frac{1}{n} \sum_{j=1}^{n} 1\{x_j \in s_{ij}\}$$

where $h_i$ is the fraction in the $i^{th}$ bin, $x_j$ is the $j^{th}$ observed value, $s_{ij}$ is the $i^{th}$ gap associated with the $j^{th}$ forecast, and

20 $1\{\cdot\}$ is a step function that gives a value of 1 if the condition is met and 0 otherwise.

The mean Continuous Ranked Probability Skill Score ($\overline{\text{CRPSS}}$) measures the performance of one forecasting system compared to another forecasting system in terms of the mean Continuous Ranked Probability Score ($\overline{\text{CRPS}}$). The $\overline{\text{CRPS}}$ measures the average square eror of a probability forecast across all possible event thresholds. The $\overline{\text{CRPSS}}$ comprises a ratio of the $\overline{\text{CRPS}}$ for the forecasting system to be evaluated $\overline{\text{CRPS}}_{\text{EVAL}}$, and the $\overline{\text{CRPS}}$ for a reference forecasting system, $\overline{\text{CRPS}}_{\text{REF}}$.

$$\overline{\text{CRPSS}} = \frac{\overline{\text{CRPS}}_{\text{REF}} - \overline{\text{CRPS}}_{\text{EVAL}}}{\overline{\text{CRPS}}_{\text{REF}}}$$

As a measure of the average square error in probability, values for the $\overline{\text{CRPS}}$ approaching zero are better. As a result, values for the $\overline{\text{CRPSS}}$ closer to one are better as this illustrates that $\overline{\text{CRPS}}_{\text{EVAL}} < \overline{\text{CRPS}}_{\text{REF}}$.



### 2.9.2 Splitting the Analysis Based on Rainfall

The qualitative analysis shown in the following Results section illustrates that there is a significant difference in the abilities of the filters to effectively project streamflow when it is raining and when it is not raining. As a result, the quantitative analysis is split into two parts: 1) for periods in which it is raining and just afterwards (for the remainder of each respective filter or projection period), and 2) for periods in which it is otherwise not raining.

Note that these periods do not necessarily correspond to the rising-limb and recession periods of the hydrograph since the river does not always respond strongly to the precipitation for the time period of study in this basin. As a result, for lack of better terminology, these periods are hereafter referred to as "rain-influenced" and "rain-free." It would be more correct to say "periods during and immediately after the rainfall within the 3-day period" and "otherwise rain-free," but that would be too cumbersome in the remainder of the text, so we ask for your indulgence in the potentially confusing use of the terms "rain-influenced" and "rain-free." Furthermore, it is also important to note that the terms "rain-influenced" and "rain-free" only refer to a time period rather than the discharge of the river. The time period that these terms refer to are the portions of the 3-day time period under consideration in the analysis.

Recall the description for Figure 2 in Section 2.7.3. for illustrating the difference between "rainfall" and "rain-free". The filter periods are rain-free for the July 22 rain event in ABC1 through to ABC12 as shown by the orange and dark green bars, while the filter periods are rain-influenced in ABC7 through to ABC12 as shown in the light green bars. Similarly, the projection periods are rain-influenced in ABC1 to ABC6 as shown by the pink bars and rain-free in ABC1 to ABC12 as shown in the red bars. In both cases (filter and projection), the rain-influenced period is considered to be from the beginning of the rainfall (July 22, 9 EST in Figure 2) to the end of the corresponding filter or projection period that "sees" the precipitation.

The beginning and ending of the precipitation events are considered as follows:

– July 22, 14 UTC (9 EST) to 20 UTC (15 EST)

– August 11, 14 UTC (9 EST) to 20 UTC (15 EST)

– September 10, 08 UTC (3 EST) to September 11, 02 UTC (September 10, 21 EST)

– September 19, 20 UTC (15 EST) to September 20, 20 UTC (15 EST)

– October 3, 08 UTC (3 EST) to 20 UTC (15 EST)

For the rain-influenced and rain-free periods, the quality of the ensemble mean, distribution and skill are compared. The skill is calculated with an unskilled reference forecast, which in this study is taken to be the measured streamflow at 00 UTC and 12 UTC each day as the forecast for the next 72 hours. This reference forecast is a persistence forecast, which assumes the streamflow is persistent for the forecast period. The June 3 rain-influenced period is not assessed due to the fact that it was not a projection period in the preceding streamflow and precipitation filter.



### 2.9.3 Verification of the H-EPS

As with the earlier analysis when precipitation uncertainty is minimized, the mean error and CRPSS are calculated for stream-flow for both rain-influenced and rain-free periods as determined by precipitation events in the basin. The overall mean error and CRPSS is also calculated.

The mean error, rank histograms and CRPS of the REPS precipitation ensemble mean are calculated using CaPA as the observation. The CRPSS is also calculated using the June to October CaPA "climatology" as the unskilled reference forecast. The mean error, CRPS and CRPSS are calculated above and below the ninetieth percentile of CaPA precipitation (0.42 mm/hr).

## 3    Results

The results are ordered according to four filter configurations and then the H-EPS. In all cases, the time period of interest is
short-term, which is defined here as 3 days.

### 3.1    Minimized Uncertainty Filter

In determining the effectiveness of the P-SEDA approach, it is necessary to see if the method has the possibility of succeeding with accurate streamflow and precipitation data. Figure 3 shows the match between the top 10 runs of each 3-day period for the months of June 1 to October 31 for the years 2003 to 2008. The black lines represent the observed daily streamflow while
the red lines represent an overlap of the top 10 matches to streamflow for every 3-day period beginning at 0 UTC and 12 UTC on each day.

These results indicate that the P-SEDA method using this particular model generally has the capability to simulate the observed streamflow for this basin if the optimal parameters are selected on each 3-day window. The notable exceptions are in early September, 2004 and much of June and July in 2007. A qualitative analysis of the actual precipitation was completed for
early June, 2007 (not shown). Historical radar images show that it is possible that CaPA underestimates the precipitation in the basin for these time periods. The presence of streamflow that is not simulated in any of the 10,000 model runs indicates that it is quite likely that CaPA does not produce enough precipitation for this specific basin in the first half of June, 2007. We suspect that the reason for any obvious mis-match between the P-SEDA results and observations is due to limitations in CaPA, which are unavoidable due to the lack of precipitation observations in the basin.
To examine the P-SEDA approach in more detail at a higher temporal resolution, the method is also applied for June 1 to October 31, 2014 and compared with hourly (rather than daily) streamflow observations. Figure 4a shows precipitation from CaPA. Figure 4b shows the observed streamflow (black) and corresponding top model runs (red). Figure 4c shows the corresponding basin-average water storage values for each of the parameter-state pairs chosen by the minimized uncertainty filter. These storage results will be discussed later.
For this study, the qualitative results in Figures 3 and 4b) illustrate that CaPA precipitation cascades to reasonable streamflow values most of the time. Since the objective of this study is to examine the effectiveness of the P-SEDA approach, we can simply




ignore time periods where CaPA clearly fails to produce the appropriate precipitation for the basin. This allows us to focus our remaining analysis on the hourly results from the beginning of June to the end of October, 2014. Focusing on this time period also has the advantage of illustrating some results using the latest version of ECCC's meteorological REPS, which was implemented on December 4, 2013.

A quantitative analysis compares these results to the other filters, but a qualitative analysis is first performed for each of the filters.

## 3.2   Bulk Calibration Filter

This filter uses the more classical hydrological approach of bulk calibration (or parameter estimation), where optimal parameter sets are based on a long time-series of streamflow. In this case, the top parameter sets are filtered based on a calibration of the

daily streamflow from June 1, 2003 to October 31, 2008, and then applied for the period of June 1 to October 31, 2014.

    A rigorous calibration is not performed in favor of using the same LHS parameter sets to sample the parameter space. Other approaches to bulk calibration would certainly prove to be more fruitful if finding an ensemble of optimized runs from bulk calibration was the goal of this research, but using the same pool of parameter sets for all of the filters explored here allows for a fair comparison of different filter techniques for the P-SEDA method.

The top 10 RMSE simulations (from the 10,000 LHS runs) for the period of June 2, 2002 to June 1, 2009 are selected from the continuous simulations for analysis for the period of June 1 to October 31, 2014, the results of which are seen in Figure 4d. The results are similar if any June to October period for 2002 to 2008 is used to determine the top 10 RMSE simulations (not shown). With the exception of early June, most calibrated runs overestimate the observed streamflow.

## 3.3   Preceding Streamflow Filter

Another manner in which to filter the parameter sets (and associated states) is to consider only the preceding streamflow. In this study, the best RMSE values from the preceding 3 days of streamflow are used to determine the parameter sets to use for the subsequent 3 days of streamflow. This process is repeated twice daily at 0 UTC and 12 UTC (19 and 5 local time for the basin in question) for June 1 to October 31, 2014.

    Figure 4e shows the overall results. Qualitatively, the filter produces good results when there is negligible precipitation.

However, the results degrade when it rains. To illustrate this aspect of the filter in more detail, Figures 5a and 5b show how the filter reacts for a single rain event on July 22, 2014. In Figure 5a, which is the equivalent of ABC6 in Figure 2, the filter period does not "see" the rain and the projected streamflows resulting from the filtered parameter sets overestimate the actual streamflow. In Figure 5b (equivalent to ABC7 in Figure 2), the rain event occurs during the filter period and the subsequent projected streamflows are much more closely aligned with the observations. This result is consistent with all

significant precipitation events, with the filter choosing parameter sets that overestimate streamflow when the precipitation event is not "seen" by the filter.





### 3.4 Parameter and Preceding Streamflow Filter

The third filter explored here is one in which the top simulations are selected based on the preceding streamflow and parameter ranges that are proven to be important during the 2014 precipitation events. Figure 6 shows parameters that are particularly sensitive during six precipitation events. Each parameter is normalized between 0 and 1. Based on these box-plots, the 10,000

parameter sets are reduced to 91 parameter sets by confining the values of the normalized parameters as follows: KS1 < 0.1, WF_R2 > 0.6, CLAY11 > 0.5, CLAY12 > 0.5, SDEP1 > 0.2. Using the preceding streamflow filter with these 91 parameter sets to obtain the top 10 runs for each 3-day period yields Figure 7. With the exception of the June 3 precipitation event, these results are clearly much better than those found in Figures 4c and 4e.

### 3.5 A Quantitative Comparison of Filters

Table 4 shows the mean error of the ensemble mean for the previously defined rain-influenced and rain-free periods for the 3-day filter and the three projection methods, and the mean error for the the unskilled forecast. Most methods, including the unskilled method, provide reasonable results for the rain-free periods. The only exception is the bulk filter, which shows a slight positive bias in the predictions. For the rain-influenced periods, which are the real periods of concern for this study, the filter is capable of finding parameter sets that have a low mean error at 24, 48 and 72 hours. The 3-day projection period

performs the worst in terms of overpredicting streamflow in rain-influenced periods, followed closely by the bulk calibration. The 3-day projection with constrained parameters performs close to the 3-day filter with only a slight over-prediction of the observed flows.

Although not shown, the rank histograms illustrate that the 3-day filter is under-dispersive for the rain-free periods and over-predicts the rain-influenced periods. The bulk filter over-predicts both periods while the 3-day filter with constrained

parameters over-predicts the rain-free periods and generally has the correct average spread for the rain-influenced periods.

Table 5 shows the skill of the filter and projection methods using the current streamflow as the unskilled forecast. The filter exhibits a very high skill for rain-free periods. Also for the rain-free periods, the 3-day projection shows some skill, while the 3-day projection with constrained parameters and the bulk projection show no skill. For the rain-influenced periods, the only projection that shows any skill is the 3-day projection with constrained parameters. These results quantify the qualitative

analysis shown in Figures 4a to 4c and 7.

### 3.6 H-EPS

To address the question of how this data assimilation approach could be used in a forecasting context, a full H-EPS is used to force selected parameter-state ensemble members with ECCC's Meteorological Regional Ensemble Prediction System (REPS), as described in the methodology section. Three sets of parameter-state ensembles are selected to see how the REPS performs.

The ensembles are based on 1) the preceding streamflow filter, 2) the parameter and preceding streamflow filter, and 3) the minimized uncertainty filter.





Figure 8a) shows CaPA (reddish brown) and the 20 REPS precipitation members (blue). The resulting 200 streamflow ensembles from each of the filters are shown in Figures 8b to d, with the grey lines coming from the preceding streamflow filter, the orange lines coming from the parameter and preceding streamflow filter, and the green lines coming from the minimized uncertainty filter. In all cases, the black line is the observed streamflow. Even for the minimized uncertainty filter, that shows

near-perfect alignment with the observed streamflow when forced with GEM and CaPA, these few members that overestimate the precipitation have an impact on the resulting ensemble of streamflows. For the preceding streamflow filter, the parameter-state pairs are completely inappropriate for making streamflow projections when it rains (note the difference in scale for flow between Figure 8b to d.)

Table 6 shows the mean error for streamflow and Table 7 shows the CRPSS, for both rain-free and rain-influenced periods.

The overall mean error and CRPSS are also calculated.

The mean error results show that the H-EPS ensemble mean overestimates streamflow in all cases. The CRPSS scores show that the H-EPS fails to show skill during key time periods for many of the ensembles when compared to using the current streamflow as the forecasted streamflow. This lack of skill will be discussed in the discussion. To examine these findings with respect to the precipitation; the mean error, rank histograms and CRPS of the REPS precipitation ensemble mean are calculated

using CaPA as the observation. The CRPSS is also calculated using the June to October CaPA "climatology" as the unskilled reference forecast. The mean error, CRPS and CRPSS shown in Table 8 are calculated above and below the ninetieth percentile of CaPA precipitation (0.42 mm/hr). Below this threshold, the REPS mean precipitation over-estimates the CaPA precipitation. In the top 10 percent of CaPA precipitation values, however, the REPS mean under-estimates the CaPA precipitation. The rank histograms (not shown) indicate that the ensemble members tend to underestimate precipitation, although some REPS

members do over-estimate the higher CaPA precipitation values. The CRPS shows the highest (worst) values, and the CRPSS shows the least skill, for the highest precipitation rates.

## 4   Discussion

The discussion is organized around three questions. The first question looks at whether-or-not the P-SEDA approach is capable of reproducing observed streamflow, which corresponds to the minimized uncertainty filter. The second question considers

the effectiveness of the remaining three filtering approaches. The third question revolves around the more realistic example of using the approach in a full H-EPS. Finally, advantages and limitations of the approach are discussed.

### 4.1   Given maximum data certainty, can the P-SEDA approach reproduce observed streamflow?

Although the P-SEDA filter could be applied to a hydrological model with few parameters, the Canadian MESH model is used with many parameters perturbed. This increases the dimensionality of the problem and is, to our knowledge, the first hydrolog-

ical application of ABC in a short-term DA application with such a parameter-intensive model. Although much simpler models tend to dominate the operational hydrological modelling community, part of the motivation behind using a hydrologically-



enhanced land-surface scheme in the case study is to begin laying some foundation for using such parameter-intensive models for operational ensemble hydrological forecasting.

One major limitation to the way in which MESH is applied in this study is the use of the relatively inefficient Latin Hypercube Sampling to determine the prior distribution of parameter sets to be used with the ABC approach. Despite this limitation,

however, the results clearly show that the approach can find parameter-state sets that match the observed hydrograph over successive periods of a few days. The exception is when the streamflow clearly shows a signal that indicates that precipitation occurred in the basin, but the model is not forced with rain. One possible way of dealing with the uncertainty in precipitation for this basin is to perturb the CaPA precipitation field as is examined by Carrera et al. (2015).

The widely varying nature of the simulated basin storage for the selected runs for each 3-day period also highlights a

limitation with the study. This limitation is in only using streamflow as the state variable to determine the top parameters each time. Consider the following water balance equation for the basin: $P - E = R + \mathrm{d}S/\mathrm{d}t$, where $P$ is precipitation, $E$ is evapotranspiration, $R$ is runoff and $\mathrm{d}S/\mathrm{d}t$ is the change in basin storage over time. Over the short time-periods of a few days in short-term hydrological prediction, $E$ can generally be ignored, leaving only $P = R + \mathrm{d}S/\mathrm{d}t$. In the hindcasting exercise presented in this part of the study, $P$ and $R$ are considered to be known and the only remaining term is $\mathrm{d}S/\mathrm{d}t$. So why does the

analysis show such a wide range of basin storage terms for the best matching assimilated streamflow? The answer lies in the fact that it is not the basin storage that balances the equation, but rather the change in storage over the time period of interest. The model is capable of releasing or storing the appropriate amount of water in both rain-influenced and rain-free scenarios, and the model determines $\mathrm{d}S/\mathrm{d}t$ based on the interaction of existing storage, model physics and parameters.

The issue of widely-varying simulated basin storage also highlights the issue of equifinality. The model is able to find

many parameter-state sets that fit the streamflow for short periods of time. However, including the state of basin storage in the assessment of equifinality clearly shows that the parameter-state sets are not equal. One assumption in most environmental modelling exercises is that the parameters do not vary with time, or at least they vary slowly or if the system is disturbed in some way such as land-use change (Bard, 1974; Wagener et al., 2003; Liu and Gupta, 2007). Wagener et al. (2003) indicate that the inability of a single parameter set to simulate an entire streamflow record provides evidence of model structural error.

It is incorrect to assume that MESH has a perfect model structure, so the results indicate that any model structural errors can be compensated for by the parameter sets. One can also presume that data errors can also be hidden by the selection of certain parameter sets. Clearly the model needs further constraints to give the results a more solid foundation. One of these constraints could be the assimilation of some aspects of storage in the model. One such possibility would be to examine the usefulness of the soil moisture and ocean salinity (SMOS) satellite (Mecklenburg et al., 2012; Jackson et al., 2012).

## 4.2  How well do different filters work?

The issues of parameter time-invariance and the most appropriate model structures are generally secondary considerations in DA. The focus in DA shifts from the exercise of improving the model and its parameterization to the exercise of making a more accurate prediction. The results presented from the various filters tested, however, indicate that some thought is required to determine the appropriate parameter sets at the appropriate times.





The only filter in this study that shows any capability in predicting streamflow when it rains is the preceding streamflow filter with constrained parameters. The manner in which this filter is applied in this study reveals that 91 of the original 10,000 LHS parameter sets can be used effectively with the P-SEDA filter approach to perform short-term predictions in the basin for the months of July to October, 2014. Techniques other than LHS must be explored to obtain more appropriate parameter sets for the P-SEDA method to work in this type of situation.

The fact that constraining the parameter sets allows for the approach to produce reasonable results throughout the period provides some assurance that the method has the possibility of being able to predict streamflow given a certain amount of precipitation. The key, at least in part, is expected to be in using a method other than LHS to determine the prior distribution of parameter sets. Alternative approaches could use algorithms such as Dynamically Dimensioned Search - Approximation of Uncertainty (DDS-AU) which have been shown to be more efficient than GLUE (Tolson and Shoemaker, 2008). The prior can also be obtained by looking for parameter sets that perform well for different hydrological signatures (e.g. Zhang et al., 2014; Shafii and Tolson, 2015) or different hydrological scenarios which might include streamflow responses to snow-melt, runoff over frozen ground, rain during wet conditions, rain during dry conditions, or whatever else can be considered a relevant hydrological event affecting streamflow.

If given more information about the state of the basin (other than streamflow), different hydrological scenarios could also be used in determining the appropriate parameter-state sets to filter. For example, if the SMOS satellite indicates that the basin is dry, the streamflow observation is relatively low and a certain amount of precipitation is expected in the near future, then past scenarios that fit this description could be used to filter the parameter sets. As a result, parameter sets that fit both the current state of the basin as well as the expected forcing could be filtered, if both the current basin state and expected precipitation has been previously experienced and observations are available.

Such an approach is very similar to the well-established $k$-nearest neighbor ($k$-nn) bootstrap method as described by Lall and Sharma (1996). In its simplest form, the $k$-nn approach finds $k$ similar patterns in the past data and uses this information to make a prediction about the next data point. The P-SEDA preceding streamflow filter essentially does the same thing, except that it looks for similar patterns in an ensemble of model runs rather than in a time series of data points. By including criteria beyond streamflow as suggested in the previous paragraph, one could (for example) look for past parameter sets that successfully simulated the streamflow when the basin exhibited a certain threshold of upper-layer soil moisture from SMOS, a given streamflow, and a specified amount of precipitation. This approach requires a relatively long time series of observational data with model simulations and could provide an interesting comparison between the model-centric P-SEDA filter and purely data-driven analogue methods.

## 4.3 How can this approach be used in a forecasting context (including precipitation uncertainty)?

The mean error results for the H-EPS ensemble mean streamflow, forced with the REPS (Table 6), are similar in nature to the mean ensemble streamflow forced with GEM and CaPA (Table 4). The H-EPS with the REPS generally performs better in rain-free periods than rain-influenced periods, and the preceding streamflow projection during rain-influenced periods performs the worst. However, an important finding is drawn from the CRPSS scores in Table 7. Overall, the two projected parameter-state





pair ensembles do not show skill, particularly for rain-influenced time periods. The H-EPS skill is lowest for the parameter and preceding streamflow filter, for both the overall and for the rain-free periods. But the lack of skill during the predominantly rain-free periods of this study is offset by the improved skill for the rain-influenced periods. In particular, the third day shows some skill for the ensembles resulting from the constrained parameters.

These findings illustrate that the H-EPS contains too much uncertainty to be used with any skill for this particular study. It is important to note that the same lack of skill may not be true for other time periods or different basins. For this particular study, it is not surprising that the REPS does not show any skill when compared to using the current streamflow as the forecast. For this basin and the time period considered, the streamflow is not very responsive to the precipitation input for much of the time. Situations when the river is not responsive to precipitation favor the approach of using the current streamflow as the forecast.

A resulting question is whether or not the lack of skill in the H-EPS is due to the uncertainty in the REPS precipitation, or the unresponsive behaviour of the streamflow to precipitation during this period. Looking more closely at the REPS precipitation mean error compared to CaPA (Table 8) indicates that the REPS tends to overestimate the bottom 90 percent, and underestimate the top 10 percent, of CaPA values, which are taken to be as close to observed as is possible in the basin. The only noticeable trend in time is that the underestimation in the top 10 percent of precipitation becomes more pronounced with time.

The unresponsive streamflow in this study is likely due to "fill-and-spill" dynamics (Spence, 2010). Being on the Precambrian Shield, and the starting point of many streams in the basin being small lakes, there are many parts of the basin that need to be filled-up before they contribute to streamflow. This physical process, especially with respect to the headwater lakes, is not represented in the version of MESH used in this study. Future work should focus on this aspect more closely.

Returning to the question of whether-or-not the lack of skill in the H-EPS is due to the uncertainty in the REPS precipitation,
or the unresponsive behaviour of the streamflow to precipitation during this period, it seems that both factors contribute to the overall lack of skill. As Figure 8, shows, however, relatively small differences in precipitation result in large changes to streamflow, indicating that the land-surface physical processes (e.g. fill-and-spill) that determine the responsiveness of the streamflow to precipitation, are probably the more important of the two for this particular study.

The only other use of ECCC's REPS as a part of an H-EPS is found in Abaza et al. (2013), in which the Canadian operational
meteorological Global Ensemble Prediction System (GEPS) was compared with the REPS and the deterministic 15 km GEM NWP forcing of the province of Quebec's operational streamflow forecasting system. The study found that both the GEPS and REPS outperformed a deterministic run for eight watersheds ranging in size from 355 to 5820 $km^2$. The REPS was also found to be superior to the GEPS in terms of its ability to predict forecast uncertainty.

One issue highlighted in the conclusion of Abaza et al. (2013) is that the REPS was found to produce unusually high pre-
cipitation spikes. This issue of excessive precipitation was, in many cases, determined to be caused by the physics perturbation scheme that was used to generate the ensemble (Erfani et al., 2014) and was fixed in the version of the REPS that was officially released on December 4, 2013. The update to the REPS is one of the main reasons for focusing on 2014 as a period of interest.





### 4.4 Advantages and Challenges of the Approach

One key benefit of the P-SEDA filter is that it is conceptually straight-forward. In plain language, the idea is to setup a series of continuous simulations and draw the most appropriate runs from these simulations for making a projection or forecast. This concept is very easy to understand and implement. In an operational forecasting environment, in which DA approaches are fundamentally designed to support, this simplicity is desirable.

Another advantage is that the parameters and state variables are always consistent with one-another. This cannot be said for other approaches such as the dual Particle Filter or dual Ensemble Kalman Filter.

As the examples provided in this study have shown, the approach is also flexible. It can be used in the more traditional manner of hydrologic model calibration, or in other unique ways that have been examined and discussed throughout the paper. It can be seen as a more general approach to model calibration, which is characterized in this study as the bulk calibration filter.

Two challenges with the approach are 1) how to determine the prior parameter sets to run in continuous simulations, and 2) how to select the most appropriate runs for making a projection or forecast. This study uses a parameter-intensive H-LSS and deals with the first challenge by using the relatively inefficient LHS to determine the prior, and deals with the second challenge by comparing three filters to select the appropriate runs. There are likely better ways of dealing with these challenges than have been explored here.

Fortunately, there is an exhaustive body of research and a number of existing tools that can be used to overcome these challenges. Possible solutions to determine a better prior include: 1) selecting parameter sets based on more than just streamflow, 2) selecting parameter sets based on different hydrological signatures or aspects of the streamflow 3) using k-nn type approach of looking for parameter sets that worked in similar circumstances in the past, 4) using more efficient algorithms than LHS to determine the prior. Any or all of these methods can be used together to improve the determination of the prior. In terms of selecting the most effective particles once the prior has been established, one method that can be explored is to use more than streamflow to select the top particles with the ABC method.

For both determining a better prior and selecting the most effective particles once the prior has been established, remote sensing offers such opportunities to gather information on the watershed state (e.g. soil moisture, snow) that can complement the limited information that streamflow provides. This approach would better constrain the model in the parameter and state estimation process, and thus help to reduce the equifinality issue. Using different hydrological signatures, or segmenting the hydrographs for different parameters (e.g. groundwater parameters during low flows), are also ideas worth exploring.

The effectiveness of these methods requires further study.

### 5 Conclusions

The main contribution of this work is the introduction of a new DA method (P-SEDA) that ensures the compatibility of parameters and states in the context of a H-EPS based on a parameter-intensive H-LSS. The DA method simplifies the traditional particle filter method by always returing to the initial particles and removing the need to resample the parameter space between





each model run. The weighting of each particle from the original set of particles is then determined using a rectangular filter, assigning each particle a value of zero or one. This simplified PF is the same as applying the ABC algorithm.

In this study, one filter is to use ABC to select the top runs from a long time-period (bulk filter), a second filter is to apply ABC for only the preceding three days of streamflow every 12 hours (preceding streamflow filter), and the third filter

is the same as the second filter with a parameter-constrained subset of the original 10,000 runs (preceding streamflow filter with parameter constraints). The parameter constraints are determined from an analysis of the filter-period results during rain-influenced periods.

The preceding streamflow filter-period results clearly show that the model and LHS method of sampling 10,000 prior parameter sets is capable of simulating the streamflow for any three-day period where the precipitation input is reasonable. The

three methods tested to select the most appropriate runs, however, show that making a projection is more complicated. The only method that consistently shows reasonable projections in this work is the preceding streamflow filter with parameter constraints. The problem with this filter is that it is not immediately clear how such a filter can be used in a forecasting context. Something more is needed to provide better parameter estimates if the P-SEDA filter is to be useful in an operational forecasting setting. Fortunately, there are a number of approaches that can be explored to provide superior guidance on the parameters,

either in pre-determining the prior or in selecting the most appropriate runs from the prior.

In addition to introducing P-SEDA, a fuller H-EPS is presented that includes forcing uncertainty from ECCC's REPS. For this particular basin and time-period, the resulting H-EPS is shown overall to be less skillful than using the current streamflow as the forecast for the future streamflow, likely due to model structural errors in MESH. This result is not generally applicable as one should expect the current streamflow to be a fairly good indicator of future streamflow when the stream is relatively

unresponsive to precipitation inputs, as is the case in this study. It is expected that the REPS precipitation in an H-EPS would exhibit more skill in more responsive basins without the same fill-and-spill physical processes or for more responsive time periods in this basin.

*Acknowledgements.* We gratefully acknowledge Environment and Climate Change Canada, the Natural Sciences and Engineering Research Council (NSERC) and Hydro Quebec for funding this study. We also thank Ethan Johnson for his editorial support and the thesis committee

for their comments on an earlier draft of this paper that is a part of the first author's PhD dissertation.





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



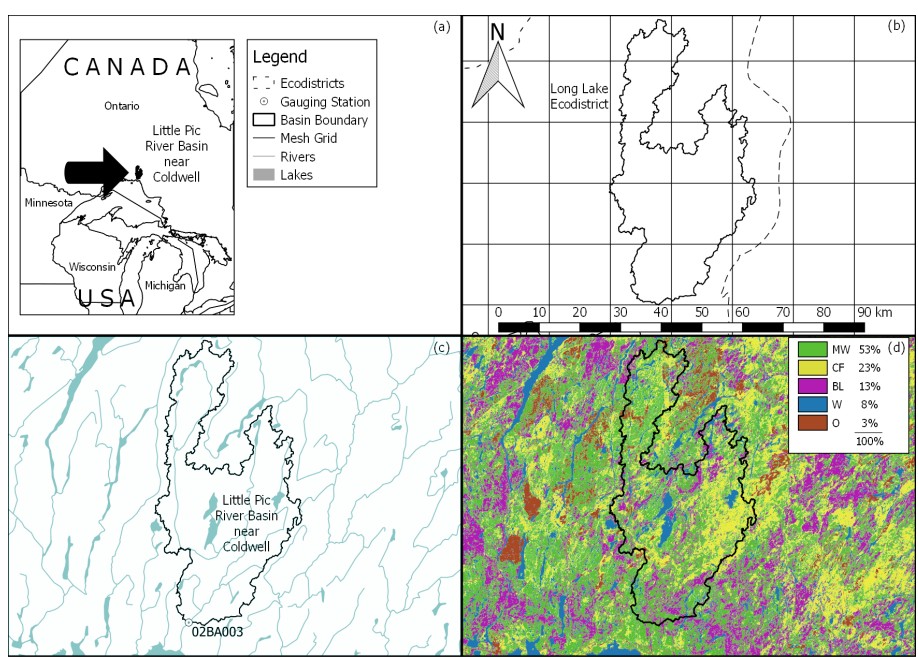

**Figure 1.** Little Pic River basin near Coldwell, Ontario, Canada. a) Location of the basin and legend, b) basin outline with respect to ecodistrict, c) river network and gauge location (02BA003), and d) landcover (MW is Mixed Wood, CF is Coniferous Forest, BL is Broadleaf Forest, W is Water, and O is other).





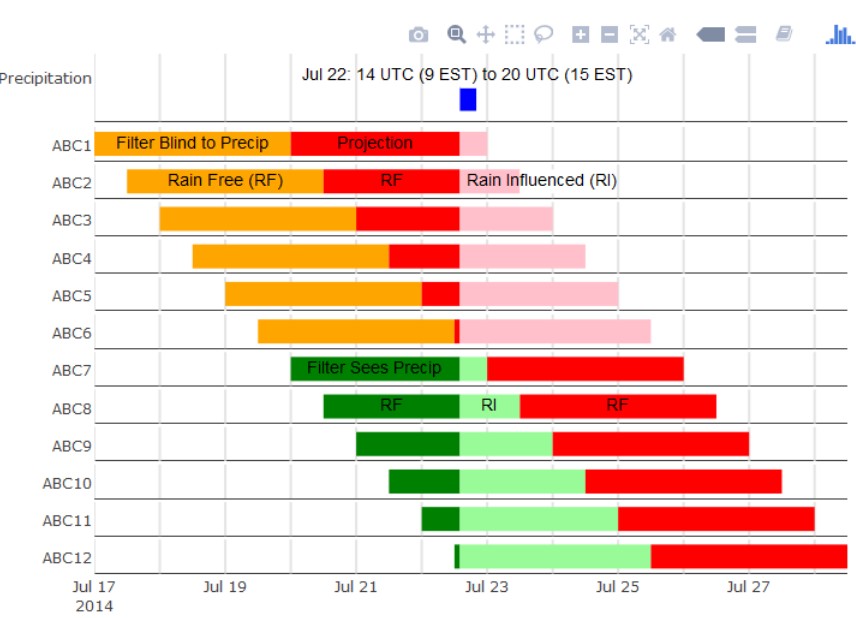

**Figure 2.** Preceding Streamflow Filter and Projection Periods for July 17 to July 28, 2014 using Hourly Streamflow. This Figure is fully explained in sections 2.7.3 and 2.9.2.







**Figure 3.** Results of the P-SEDA filter for multiple 3-day filter periods. The black lines represent the observed daily streamflow while the red lines represent an overlap of the top 10 matches to streamflow for every 3-day period beginning at 0 UTC and 12 UTC on each day. Where only a black line is seen, it simply covers the red lines completely.



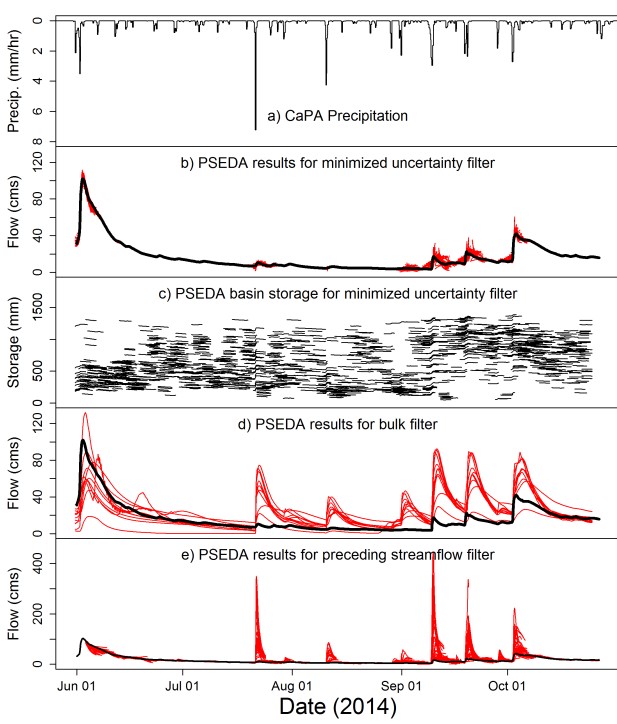

**Figure 4.** CaPA precipitation (a) for all simulations shown in this Figure. (b) Observed streamflow (black) and top 10 streamflow values (red) for the minimized uncertainty filter. (c) Corresponding basin-wide storage values for the minimized uncertainty filter. (d) Results of the bulk calibration filter. (e) Top 10 preceding streamflow filter projections in each of the 3-day filter periods. The black lines in (b), (d) and (e) show observed streamflow with different y-axis scaling.





**Figure 5.** A single filter-projection period for two neighboring time periods. For the two sub-plots in a) the projection begins at 7:00 local time, July 22, 2014 (12 UTC). For the two sub-plots in b) the projection begins at 19:00 local time, July 22, 2014 (0 UTC, July 23). The upper plot of each sub-figure shows CaPA precipitation and instantaneous storage. The lower plot shows observed streamflow (black), the top 10 runs for the filter period (blue), and the corresponding streamflow projections (red).



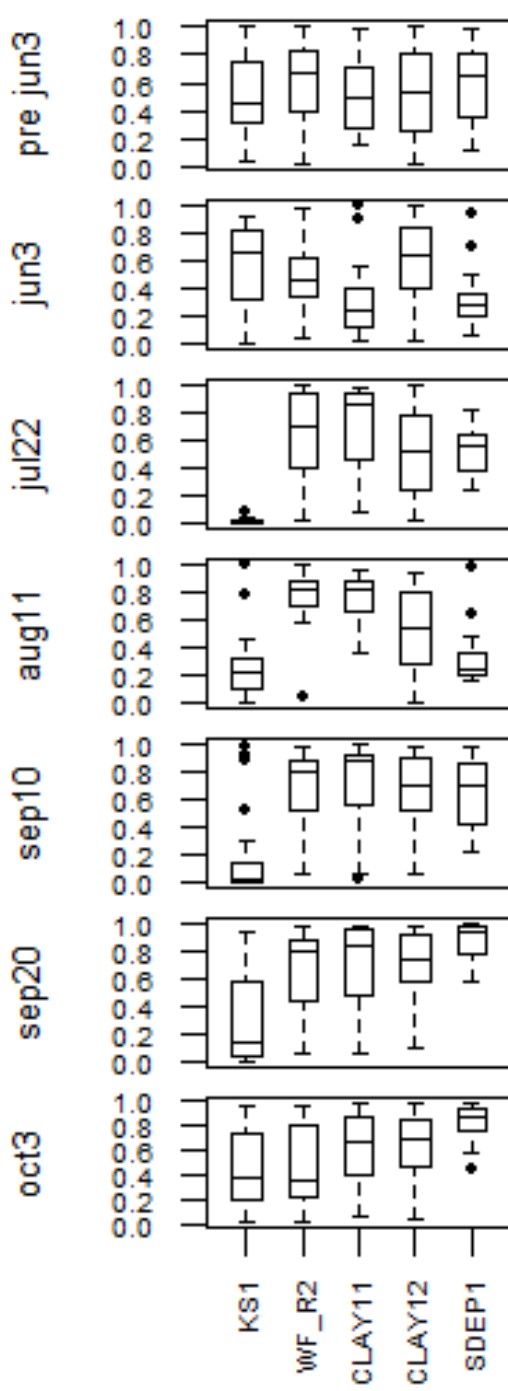

**Figure 6.** Importance of (normalized) parameters that see precipitation. The top set of box-plots shows that none of the top parameter sets have identifiable parameter values prior to the June 3, 2014 precipitation event. This result is similar to all parameter sets immediately prior to precipitation events that do not see the events. The remainder sets of box-plots show the parameter ranges for the top simulations during precipitation events.





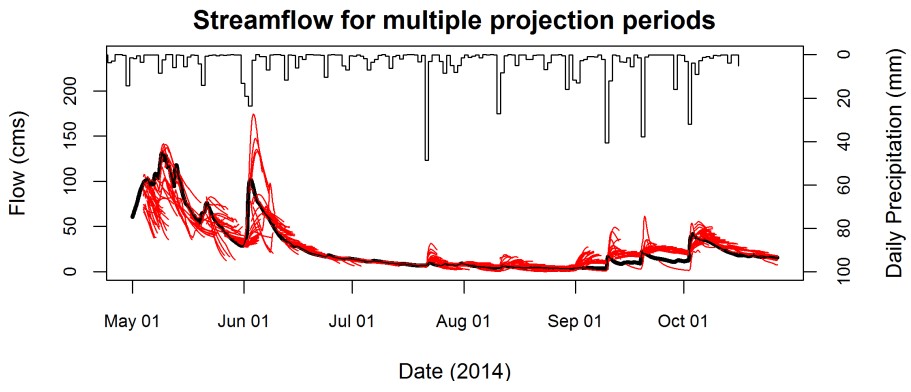

**Figure 7.** Projection period results after filtering based on parameter values and preceding streamflow.





**Figure 8.** Results of the H-EPS for the 3-day period beginning at 6 Eastern Standard Time (EST) on July 21, 2014. The reddish brown line in sub-figure a) is the Canadian Precipitation Analysis (CaPA) while the blue lines represent the 20 Regional Ensemble Prediction System (REPS) precipitation traces. The single black line in each of the sub-figures b), c) and d) are the observed streamflow. The grey, orange and green lines show the 200 H-EPS streamflow traces for the projection periods of the preceding streamflow filter, parameter and preceding streamflow filter, and the minimized uncertainty filter, respectively. Note the difference in scale for streamflow between subfigures b, c and d.



**Table 1.** Landcover percentages based on LCC2000-V Landsat product.

| Landcover | Percentage |
|---|---|
| Water | 8 |
| Coniferous Dense Forest | 23 |
| Broadleaf Dense Forest | 13 |
| Mixed Wood | 53 |
| Other | 3 |





**Table 2.** Fixed landcover parameters.

| Parameter Name | Description | Units | Value | Source |
|---|---|---|---|---|
| QA50 - NL | Reference value of incoming shortwave radiation used in stomatal resistance formula (Needleleaf) | $[\text{W m}^{-2}]$ | 30 | Verseghy (2011) |
| QA50 - BL | Reference value of incoming shortwave radiation used in stomatal resistance formula (Broadleaf) | $[\text{W m}^{-2}]$ | 40 | Verseghy (2011) |
| VPDA - NL | Vapour pressure deficit coefficient used in stomatal resistance formula (Needleleaf) | [ ] | 0.65 | Verseghy (2011) |
| VPDA - BL | Vapour pressure deficit coefficient used in stomatal resistance formula (Broadleaf) | [ ] | 0.5 | Verseghy (2011) |
| VPDB - NL | Vapour pressure deficit coefficient used in stomatal resistance formula (Needleleaf) | [ ] | 1.05 | Verseghy (2011) |
| VPDB - BL | Vapour pressure deficit coefficient used in stomatal resistance formula (Broadleaf) | [ ] | 0.6 | Verseghy (2011) |
| PSGA - NL | Soil moisture suction coefficient used in stomatal resistance formula (Needleleaf) | [ ] | 100 | Verseghy (2011) |
| PSGA - BL | Soil moisture suction coefficient used in stomatal resistance formula (Broadleaf) | [ ] | 100 | Verseghy (2011) |
| PSGB - NL | Soil moisture suction coefficient used in stomatal resistance formula (Needleleaf) | [ ] | 5 | Verseghy (2011) |
| PSGB - BL | Soil moisture suction coefficient used in stomatal resistance formula (Broadleaf) | [ ] | 5 | Verseghy (2011) |
| ROOT - NL | Root depth (Needleleaf) | [m] | 0.05 | User Selected |
| ROOT - BL | Root depth (Broadleaf) | [m] | 0.05 | User Selected |
| DDEN | Drainage density, equal to the length of the stream divided by area drained by the stream (Basin wide) | $[\text{km km}^{-2}]$ | 50 | Dingman (2002) |
| XSLP | Average overland slope. | [rise/run] | grid-based | Calculated from Digital Elevation Model |
| GRKF | Ratio of saturated horizontal hydraulic conductivity at a depth of 1 metre to the saturated horizontal hydraulic conductivity at the surface (Basin wide) | [ ] | 0.01 | User defined |



**Table 3.** Ranges for the perturbed parameters.

| Parameter Name | Description | Units | Lower Limit | Upper Limit | Source |
|---|---|---|---|---|---|
| MANN | Manning's n for overland flow. | $[\mathrm{m\ s^{-1/3}}]$ | 0.02 | 0.16 | Dingman (2002) |
| KS | Saturated surface horizontal soil conductivity. | $[\mathrm{m\ s^{-1}}]$ | 0.00001 | 0.1 | User specified |
| ZSNL | Limiting snow depth below which coverage is less than one-hundred percent. | $[\mathrm{m}]$ | 0.1 | 1 | User specified |
| SDEP | Soil permeable depth, set to greater than model soil depth to simulate fully permeable soil. | $[\mathrm{m}]$ | 0.1 | 4.2 | User specified |
| WF-R2 | River roughness factor that incorporates a channel shape and width to depth ratio as well as Manning's n. | $[\mathrm{m^{0.5}\ s^{-1}}]$ | 0.3 | 1 | User specified |
| RSMN-NL | Minimum stomatal resistance (Needleleaf) | $[\mathrm{s\ m^{-1}}]$ | 175 | 225 | Verseghy (2011) |
| RSMN-BL | Minimum stomatal resistance (Broadleaf) | $[\mathrm{s\ m^{-1}}]$ | 100 | 150 | Verseghy (2011) |
| SAND-L1 | Sand in soil layer 1. | [%] | 35 | 58 | Ecodistrict based |
| SAND-L2 | Sand in soil layer 2. | [%] | 35 | 58 | Ecodistrict based |
| SAND-L3 | Sand in soil layer 3. | [%] | 35 | 58 | Ecodistrict based |
| CLAY-L1 | Clay in soil layer 1. | [%] | 0 | 37 | Ecodistrict based |
| CLAY-L2 | Clay in soil layer 2. | [%] | 0 | 37 | Ecodistrict based |
| CLAY-L3 | Clay in soil layer 3. | [%] | 0 | 37 | Ecodistrict based |
| LANZ0-NL | Natural log of roughness length (Needleleaf). | $[\ln(\mathrm{m})]$ | -0.7 | 1.1 | Verseghy (2011) |
| LANZ0-BL | Natural log of roughness length (Broadleaf). | $[\ln(\mathrm{m})]$ | -0.7 | 1.1 | Verseghy (2011) |
| ALVC-NL | Visible albedo (Needleleaf). | [ ] | 0.02 | 0.09 | Verseghy (2011) |
| ALVC-BL | Visible albedo (Broadleaf). | [ ] | 0.02 | 0.09 | Verseghy (2011) |
| ALIC-NL | Near infrared albedo (Needleleaf). | [ ] | 0.1 | 0.5 | Verseghy (2011) |
| ALIC-BL | Near infrared albedo (Broadleaf). | [ ] | 0.1 | 0.5 | Verseghy (2011) |
| LAMAX-NL | Maximum leaf area index (Needleleaf). | [ ] | 1.8 | 2.2 | Verseghy (2011) |
| LAMAX-BL | Maximum leaf area index (Broadleaf). | [ ] | 4 | 10 | Verseghy (2011) |
| LAMIN-NL | Minimum leaf area index (Needleleaf). | [ ] | 1.4 | 1.8 | Verseghy (2011) |
| LAMIN-BL | Minimum leaf area index (Broadleaf). | [ ] | 0.2 | 4 | Verseghy (2011) |
| MAXMASS-NL | Standing biomass density (Needleleaf). | $[\mathrm{kg\ m^{-2}}]$ | 5 | 40 | Verseghy (2011) |
| MAXMASS-BL | Standing biomass density (Broadleaf). | $[\mathrm{kg\ m^{-2}}]$ | 5 | 40 | Verseghy (2011) |
| ZPLS | Maximum water ponding depth for snow-covered areas. | $[\mathrm{m}]$ | 0.1 | 0.5 | User specified |
| ZPLG | Maximum water ponding depth for snow-free areas. | $[\mathrm{m}]$ | 0.1 | 0.5 | User specified |
| DRN | Drainage index, set to 1.0 to allow the soil physics to model drainage or to a value between 0.0 and 1.0 to impede drainage. | $[\mathrm{m}]$ | 0 | 1 | User specified |





**Table 4.** Mean error ( $m^3$ $s^{-1}$) as an assessement of the ensemble mean streamflow from the P-SEDA filter (10 members) for rain-influenced and rain-free periods from June to October, 2014.

|  | Rain Free | | | Rain Influenced | | |
|---|---|---|---|---|---|---|
|  | Forecast Hour | | | Forecast Hour | | |
|  | 24 | 48 | 72 | 24 | 48 | 72 |
| unskilled | 0 | 1 | 2 | -5 | -7 | -7 |
| 3-day filter | 0 | 0 | 0 | 0 | 0 | 3 |
| 3-day projection | 1 | 1 | 1 | 61 | 53 | 42 |
| bulk projection | 6 | 6 | 5 | 25 | 30 | 31 |
| 3-day projection with constrained parameters | 2 | 2 | 2 | 1 | 3 | 3 |



**Table 5.** Mean continuous ranked probability skill score (CRPSS) as an assessement of the ensemble skill from the P-SEDA filter for rain-influenced and rain-free periods from June to October, 2014. The reference low-skill forecast is the measured streamflow at 00 UTC and 12 UTC each day as the forecast for the next 72 hours.

|  | Rain Free | | | Rain Influenced | | |
| --- | --- | --- | --- | --- | --- | --- |
|  | Forecast Hour | | | Forecast Hour | | |
|  | 24 | 48 | 72 | 24 | 48 | 72 |
| 3-day filter | 0.85 | 0.91 | 0.89 | 0.70 | 0.82 | 0.70 |
| 3-day projection | 0.07 | 0.09 | 0.16 | -6.36 | -3.57 | -2.66 |
| bulk projection | -3.97 | -1.95 | -1.06 | -1.90 | -1.71 | -1.89 |
| 3-day projection with constrained parameters | -1.06 | -0.60 | -0.39 | 0.30 | 0.44 | 0.42 |



**Table 6.** Mean error (mean H-EPS streamflow - observed) ( $m^3 s^{-1}$ ) as an assessement of the ensemble mean streamflow from the H-EPS (200 members) for rain-influenced, rain-free and overall periods from June to October, 2014.

| | Rain Free | | | Rain Influenced | | | Overall | | |
|---|---|---|---|---|---|---|---|---|---|
| | Forecast Hour | | | Forecast Hour | | | Forecast Hour | | |
| | 24 | 48 | 72 | 24 | 48 | 72 | 24 | 48 | 72 |
| 3-day filter | 1 | 2 | 3 | 1 | 2 | 7 | 1 | 2 | 3 |
| 3-day projection | 1 | 2 | 3 | 34 | 36 | 34 | 3 | 5 | 7 |
| 3-day projection with constrained parameters | 2 | 3 | 3 | 1 | 2 | 4 | 2 | 3 | 3 |



**Table 7.** Mean continuous ranked probability skill score (CRPSS) as an assessement of the ensemble skill from the H-EPS for rain-influenced, rain-free and overall periods from June to October, 2014. The reference low-skill forecast is the measured streamflow at 00 UTC and 12 UTC each day as the forecast for the next 72 hours.

| | Rain Free | | | Rain Influenced | | | Overall | | |
|---|---|---|---|---|---|---|---|---|---|
| | Forecast Hour | | | Forecast Hour | | | Forecast Hour | | |
| | 24 | 48 | 72 | 24 | 48 | 72 | 24 | 48 | 72 |
| 3-day filter | 0.77 | 0.76 | 0.67 | -0.33 | 0.12 | 0.01 | 0.71 | 0.70 | 0.60 |
| 3-day projection | 0.21 | 0.29 | 0.31 | -8.33 | -4.15 | -2.44 | -0.23 | -0.09 | -0.02 |
| 3-day projection with constrained parameters | -0.77 | -0.31 | -0.14 | -1.2 | -0.15 | 0.11 | -0.80 | -0.29 | -0.11 |





**Table 8.** Mean error (mean REPS precipitation - CaPA), CRPS and CRPSS (with JJASO, 2014 "climatology" as reference forecast) for June 1 to October 31, 2014.

|  | Mean Error | | | CRPS | | | CRPSS | | |
|---|---|---|---|---|---|---|---|---|---|
| Threshold | Forecast Hour | | | Forecast Hour | | | Forecast Hour | | |
|  | 24 | 48 | 72 | 24 | 48 | 72 | 24 | 48 | 72 |
| $Pr \leq 0.9(0.42 \text{ mm h}^{-1})$ | 0.05 | 0.06 | 0.09 | 0.03 | 0.03 | 0.04 | 0.30 | 0.19 | 0.08 |
| $Pr > 0.9(0.42 \text{ mm h}^{-1})$ | -0.14 | -0.21 | -0.19 | 0.45 | 0.49 | 0.56 | 0.38 | 0.32 | 0.22 |
| all | 0.03 | 0.03 | 0.06 | 0.08 | 0.08 | 0.10 | 0.36 | 0.28 | 0.18 |