# Peer review of "Parameter-state ensemble data assimilation using Approximate Bayesian Computing for short-term hydrological prediction"

_Hydrology and Earth System Sciences, 2017_

## Referee Comment (RC1) · Anonymous Referee #1 · 20 Nov 2017

Overall recommendation
This paper is well-written and overall very clear. The concept of applying a simple data assimilation technique to account for variations in parameters over time is interesting. The simplicity of the method is unique. It is easy to see how all versions of P-SEDA could be applied and interpreted operationally, except for constraining the parameter and preceding streamflow feature. This paper is close to being publishable as-is. My concerns are minor and can be addressed with writing.

Major concerns
1) I had to go back to the methods sections several times to interpret what methods

were covered by "P-SEDA" and how the method was applied. Algorithm 1 is extremely helpful, but I would suggest a second algorithm or a flow chart that is specific to the P-SEDA method applied, and perhaps specific to the 3-day moving window application presented in the manuscript. The manuscript should be more explicit about how P-SEDA is different from Algorithm 1.

2) I cannot tell from the description in the methods section if the filters are applied sequentially or if they always draw from the same set of 10,000 simulations. This is stated more clearly in the conclusions. Because the majority of particle filters are applied sequentially, it should be clear early on that this is not the case in this paper.

3) I would not expect three days of streamflow to be enough to determine reasonable streamflow parameters. The parameters that produce good baseflow are rarely the same parameters that produce good flood peaks. Please provide more justification for testing this method.

Minor concerns

1) Tables 4 5, lines 10-25, p. 13, and Figure 8, Conclusions: terminology suddenly changes. "Projection methods" were never defined; previously the four P-SEDA methods were all referred to as "filters". It is unclear which is the "3-day filter" and which are projections. I assume they correspond to the previously defined filters as follows, but I am not certain:

a. 3-day filter = minimized uncertainty filter

b. 3-day projection = preceding streamflow filter

c. bulk projection = bulk calibration filter

d. 3-day projection with constrained parameters = parameter and preceding stream-flow filter.

2) Line 21, p. 6: Instead of saying "ensemble data assimilation filters", say "P-SEDA" filters. Otherwise, it is not clear that P-SEDA encompasses all 4 filter approaches.

3) Lines 30-31, p. 13: I assume that you do not use the bulk calibration filter here because it performed poorly, but it is probably worth stating that.

4) Line 32, p. 17: The reason you chose to focus on 2014 should be stated in methods.

5) How do the authors propose to implement the parameter and preceding streamflow filter? Operationally, one would not extract parameters for all days during a year's precipitation events before setting the parameter range for this filter. Would it be based on the previous year's filter or would the parameter prior distribution be updated based on the days leading up to the current date? How would that impact the result? This is touched on in the conclusions, and it is probably beyond the scope of this study, but it would be helpful if this limitation were mentioned in the methods section.

6) Line 8, p. 19: Shouldn't it be the minimized uncertainty filter that shows the model is capable of simulating streamflow for any 3-day period.

Typos, grammar, etc.

1) Line 23, p. 9: "eror" -> "error"

2) Line 25, p. 13: I'm pretty sure you mean 4c to 4e and 7.

3) Lines 20-21, p. 15: "...including the state of basin storage in the assessment of equifinality clearly shows that the parameter-state sets are not equal." I'm not sure what is meant by "assessment of equifinality." Also, please refer either to a figure, a table or a citation that supports this.

4) Figure 1: Color of lakes and rivers in legend should match their color in Fig. 1c.

5) Figure 2: The toolbar at the top should be removed.

6) Figure 5: Plot storage on the same scale in a and b. I'm guessing the solid black line is precipitation and the dots are storage simulated by the best 10 parameter sets. A legend would help.

7) Table 4: "assessement" -> "assessment"

---

## Referee Comment (RC2) · Anonymous Referee #2 · 27 Nov 2017

General comments

This paper introduces a data assimilation approach that uses a filtering method to select an ensemble of best parameter sets over a certain time window from a total population of generated parameter sets. The simplicity of the method is appealing. However, several methodological choices are questionable and not properly justified.

1. The authors state that a drawback of traditional filtering methods that update the model state only or update state and parameters simultaneously provide state and parameter values that are not consistent (described in the Introduction, and re-iterated in the Discussion). But this "inconsistency" is the exact nature of filtering-based data

assimilation as opposed to variational approaches and smoothing methods. The objective of filtering is to provide the best estimate of the state (or state and parameters) at a certain time, taking the underlying model predictive and observational uncertainties into account. In the approach presented in the paper, only model parameter uncertainty is accounted for, and the results clearly show that this is not sufficient to produce skilful predictions from the filtering. These results are not surprising, since the filter does not explicitly account for one of the major error sources in hydrological modelling, the error in the precipitation forcing. Data assimilation methods that update the model state directly account for this error.

2. The filtering approach developed uses a 3-day window to select the top 10 best parameter sets. The use of a 3-day window is not justified, and it seems questionable whether such a short window is sufficient considering the different time scales of runoff responses, ranging from slowly varying baseflow to fast responding overland flow contribution. The optimal window size will depend on flow regime.

3. A latin hypercube sampling approach is applied for generating the population of parameter sets from which the top 10 parameter sets are selected in the filtering approach. The authors discuss the limitation of the LHS approach. I wonder why this limitation has not been addressed in the work. The results of the bulk calibration filter that corresponds to a classical calibration-validation approach clearly show the limitation of the LHS approach.

Detailed comments

1. Page 2, line 19-21. Not clear exactly what you mean by this statement (see General comments above).

2. Page 5, line 17-18. Explain "CLASS tile" and "GRU".

3. Page 6, line 1-2. How were the parameters and parameter intervals chosen for the LHS sampling? Based on a preliminary sensitivity analysis?

[Figure]

4. Page 6, line 11. Abbreviation "H-EPS" not defined.

5. Page 6, line 19-20. How did you justify that the choice of the 10 best parameter sets is optimal?

6. Page 7, line 9. Abbreviation "CaPA" not defined.

7. Page 10, line 8-13. A long explanation. Rephrase.

8. Page 10, line 20-25. Include a paragraph where you introduce the test period and test events.

9. Page 10, line 26-29. Description of the reference forecast is out of place here. Move to the previous section where it is already introduced (page 9, line 5-6).

10. Page 11, line 27-29. Not clear how the water storage value is calculated. Is it a state variable in the model? Or is it assessed using the water balance calculations described in the discussion?

11. Page 13, line 4-6. Not clear how the 91 parameter sets are chosen. And how can this approach be applied in an operational setting?

12. Page 13, line 11 and line 21. Use "reference forecast" instead of "unskilled forecast".

13. Page 15, line 28-29. An example of using SMOS for DA in a hydrological model can be found in Ridler et al. (2014).

14. Page 16, line 8-10. The use of LHS is identified as one of the key limitations of the approach developed. So why wasn't this issue further investigated (see General comments above)?

15. Page 18, line 6-7. Why is it an advantage that parameters and state variables are consistent (see General comments above)?

16. Page 18, line 12. Abbreviation "H-LSS" not defined.

17. Tables 2-3. Very detailed information, and difficult to understand without knowledge of the model applied. I suggest to move this to Supplementary material together with a brief description of the model applied.

18. Table 5, caption. Delete "low-skill".

References

Ridler, M.E., Madsen, H., Stisen, S., Bircher, S., Fensholt, R., 2014, Assimilation of SMOS derived soil moisture in a fully integrated hydrological and soil-vegetation-atmosphere transfer model in Western Denmark, Water Resources Research, 50(11), DOI: 10.1002/2014WR015392.

---

## Referee Comment (RC3) · Anonymous Referee #3 · 5 Dec 2017

In this paper the authors present the application of Approximate Bayesian Computation (ABC) to joint parameter and state estimation of a watershed model using a simplified particle filter. The authors consider various case studies with known/unknown streamflow and/or known/unknown precipitation. The paper presents interesting results, nevertheless, I think it requires a major revision before it can be judged to making a significant contribution to the field of hydrological data assimilation. Main issue I have with the paper is that I cannot always follow the implementation the authors are using. I will list some examples of this in my comments below. This lack of clarity downplays readability and makes it difficult to understand what has been done, and comprehend the results. I also have some more theoretical questions regarding the application of

[Figure]

ABC.

COMMENTS:

1. Section 2.1: The first few sentences describe briefly how the P-SEDA filter works. States and parameters are drawn from some multivariate initial distribution - and then analyzed for use in a projection period. How is this analysis done? I think a Figure may really help to communicate to readers how the P-SEDA method is implemented. The two sentences, "The analysis is completed and the process repeated for the next appropriate time-step in the continuous simulations" and "In this manner, both the parameters and states are drawn from the entire M simulations for the projection period". Not clear to me.

2. Line 10: so you are talking here about the normalized weights. A particle filter uses three different weights: incremental weight (for current datum only), unnormalized weight (normalized weight prior to datum x incremental weight -> summarizes weight of entire trajectory) and normalized weights -> normalization of unnormalized weights before moving on to the next datum.

3. Line 12: Resampling is the crux to an efficient implementation of the particle filter. Otherwise, many trajectories will receive a negligible weight and the PF does not approximate closely the target PDF.

4. Line 14 - 16 "The approach presented here is the same, but without resampling and always returning to the original particles as updated by the model and assigning a weight of zero or one to each particle based on the filter (i.e. using a rectangular filter)." is unclear to me. This goes back to my earlier comment. From what is presented, I do not understand how the authors implement such approach. Thus, no resampling is done? How do you return to the original particles. As with comment 1 above, can you give a detailed example, in text or in Figure that explains how this works. For example, at a time, t, we have the state forecast and associated parameter values + an incoming observation. What does the filter do then? How does it return to the original particles?

How are the weigts assigned? How is resampling avoided? etc.

5. Algorithm 1: How is $s(y\_i)$ computed? And how do you find the theta's from the k_m nearest neighbors of s0?

6. I do not understand where ABC comes in. Is this in the selection of s0? and the s_y's? And how is the likelihood function formulated? This is done by simulation, yet, I miss the details necessary to understand and comprehend what has exactly been done.

7. Latin Hypercube sampling is argued as being highly inefficient. That is true if you want to approximate a target PDF, nevertheless, if you just want to sample the parameter space, then this may be one of the best methods you can use.

8. In Section 2.7 the authors describe how they construct the ensemble. None of the four approaches listed are described in detail. Hence, I do not understand what is being done. "minimized uncertainty filter". Need a detailed explanation, step by step how we go about initial states and parameters to a minimized uncertainty filter. Same holds for the other three listed methods. Without this the results in this paper will not be understood, nor are impossible to be reconstructed by the reader.

I'll leave it with this for now. I believe the authors should clarify their methodology. Otherwise the results of this paper cannot be understood by a large audience. This would be unfortunate, as what the authors are doing has lots of potential.

---

## Author Comment (AC1) · 18 Feb 2018

Responses to RC1

Thank you for the positive "overall recommendation." In answer to your major and minor concerns:

Major Concerns

1) I had to go back to the methods sections several times to interpret what methods were covered by "P-SEDA" and how the method was applied. Algorithm 1 is extremely helpful, but I would suggest a second algorithm or a flow chart that is specific to the

P-SEDA method applied, and perhaps specific to the 3-day moving window application presented in the manuscript. The manuscript should be more explicit about how P-SEDA is different from Algorithm 1.

Author Response: A second algorithm is added in section 2.7 to illustrate how the P-SEDA method is applied and how this is different than Algorithm 1.

Pseudo-code of the implementation of the ABC algorithm in this paper
**Require:** A positive integer ($M = 10,000$) and an integer between $1$ and $M$ ($k_M = 10$).
   **for** $i = 1$ to $10,000$ **do**
      generate parameter set $i$ from all possible parameter sets using LHS;
      generate streamflow $i$ using the model (MESH)
   **end for**
   **return** The parameter sets with the lowest $10$ RMSE values between observed and simulated streamflow values from the $10,000$ model runs. (This is the filter.)

2) I cannot tell from the description in the methods section if the filters are applied sequentially or if they always draw from the same set of 10,000 simulations. This is stated more clearly in the conclusions. Because the majority of particle filters are applied sequentially, it should be clear early on that this is not the case in this paper.

Author Response: The following sentence will be added at the end of section 2.1 to clearly state up front that the filters are always drawn from the same set of parameters. "Unlike the majority of particle filters, which are applied sequentially, the P-SEDA approach always draws from the same set of $M$ simulations."

3) I would not expect three days of streamflow to be enough to determine reasonable streamflow parameters. The parameters that produce good baseflow are rarely the same parameters that produce good flood peaks. Please provide more justification for testing this method

Author Response: This is an excellent point. The relatively small size of the basin

(1,324 $km^2$) provides some justification for smaller time periods to be used as filters. The 3-day filter would likely be completely unsuitable for basins that are much larger. In addition, the results of the parameter and preceding streamflow filter illustrate that obtaining reasonable results during rainfall events require more than the 3-day filter of streamflow alone. The reviewer's point that parameters that produce good baseflow are rarely the same parameters that produce good flood peaks provides justification for an application of the P-SEDA method that includes parameter sets that produce good baseflow as well as parameter sets that produce good flood peaks, or more likely parameter sets that produce simulations that capture the transition between low and high flows.

The final sentence of the third paragraph in section 4.2 will be changed to include "... rain during dry conditions, baseflow conditions, flood peaks, transitions between low and high flows, or whatever else..."

In addition, the following paragraph after the third paragraph in section 4.2 will also be added: "The relatively small size of the basin (1,324 $km^2$) provides some justification for the relatively small time period of three days to be used as a filter. The 3-day filter would likely be completely unsuitable for basins that are much larger. However, the results of the parameter and preceding streamflow filter illustrate that obtaining reasonable results during rainfall events requires more than the 3-day filter of preceding streamflow alone. One possible solution to this problem is to have a longer filter period, but other options also exist, as described in the following paragraphs."

Finally, a short paragraph will be added at the end of section 4.2. "All such possible methods to improve the P-SEDA approach require further examination that is beyond the scope of this paper."

Minor concerns

1) Tables 4 5, lines 10-25, p. 13, and Figure 8, Conclusions: terminology suddenly changes. "Projection methods" were never defined; previously the four P-SEDA methods were all referred to as "filters". It is unclear which is the "3-day filter" and which are projections. I assume they correspond to the previously defined filters as follows, but I am not certain: a. 3-day filter = minimized uncertainty filter b. 3-day projection = preceding streamflow filter c. bulk projection = bulk calibration filter d. 3-day projection with constrained parameters = parameter and preceding streamflow filter.

Author Response: We appreciate the reviewer pointing this out and will change tables 4, 5, lines 10-25, p.13 and Figure 8 to correspond to the previous definitions.

2) Line 21, p. 6: Instead of saying "ensemble data assimilation filters", say "P-SEDA" filters. Otherwise, it is not clear that P-SEDA encompasses all 4 filter approaches.

Author Response: We agree and will make the change in text as suggested.

3) Lines 30-31, p. 13: I assume that you do not use the bulk calibration filter here because it performed poorly, but it is probably worth stating that.

Author Response: Yes. This is correct. The following sentence will be added to the end of the first paragraph in section 3.6. "The bulk calibration filter is not considered due to its poor performance."

4) Line 32, p. 17: The reason you chose to focus on 2014 should be stated in methods.

Author Response: The last two paragraphs of section 4.3 will be moved to the end of section 2.8 in the methods.

5) How do the authors propose to implement the parameter and preceding streamflow filter? Operationally, one would not extract parameters for all days during a year's precipitation events before setting the parameter range for this filter. Would it be based on the previous year's filter or would the parameter prior distribution be updated based on the days leading up to the current date? How would that impact the result? This is touched on in the conclusions, and it is probably beyond the scope of this study, but it would be helpful if this limitation were mentioned in the methods section.

Author Response: Yes. The implementation of this filter in an operational setting would be a problem. It was mainly included to show that a small subset of the 10,000 LHS runs could be used more effectively than the full 10,000 parameter sets. The application of the approach operationally would have to use a different method to determine a more effective set of parameters. As per the reviewer's suggestion, the following sentence will be added at the end of section 2.7.4: "It is worth noting here that this filter would be difficult or impossible to implement operationally for this basin in the same way it is implemented in this study, particularly in the spring and early summer. It is included in this study to highlight some of the limitations of the other filters."

6) Line 8, p. 19: Shouldn't it be the minimized uncertainty filter that shows the model is capable of simulating streamflow for any 3-day period.

Author Response: Yes. This will be corrected in the text.

Typos, grammar, etc.

1) Line 23, p. 9: "eror" -> "error"

Author response: This will be corrected.

2) Line 25, p. 13: I'm pretty sure you mean 4c to 4e and 7.

Author response: This will be corrected.

3) Lines 20-21, p. 15: "...including the state of basin storage in the assessment of equifinality clearly shows that the parameter-state sets are not equal." I'm not sure what is meant by "assessment of equifinality." Also, please refer either to a figure, a table or a citation that supports this.

Author response: Thank you for pointing this out. This sentence is not clear and the paragraph will be adjusted as follows:

"The issue of widely-varying simulated basin storage (Figure 4c) also highlights the issue of equifinality, which is defined here as the idea that many different model simulations can produce acceptable results (Beven, 1993). The model is able to find many parameter-state sets that fit the streamflow for short periods of time. If only streamflow observations are available, the selected simulations are equifinal. However, including the state of basin storage clearly shows that the parameter-state sets are not equal. If soil moisture observations are also available and used, then these simulations are not equifinal and the selected simulations can be further constrained."

The following sentence in the original manuscript will begin as a new paragraph.

4) Figure 1: Color of lakes and rivers in legend should match their color in Fig. 1c.

Author Response: The figure will be changed accordingly.

5) Figure 2: The toolbar at the top should be removed.

Author Response: The figure will be changed accordingly.

6) Figure 5: Plot storage on the same scale in a and b. I'm guessing the solid black line is precipitation and the dots are storage simulated by the best 10 parameter sets. A legend would help.

Author response: The figure will be changed accordingly.

7) Table 4: "assessement" -> "assessment"

Author response: This will be corrected.

Beven, Keith. "Prophecy, reality and uncertainty in distributed hydrological modelling." Advances in water resources 16.1 (1993): 41-51.

---

## Author Comment (AC2) · 18 Feb 2018

Responses to RC2

Major Concerns

1. The authors state that a drawback of traditional filtering methods that update the model state only or update state and parameters simultaneously provide state and parameter values that are not consistent (described in the Introduction, and re-iterated in the Discussion). But this "inconsistency" is the exact nature of filtering-based data assimilation as opposed to variational approaches and smoothing methods. The objec-

tive of filtering is to provide the best estimate of the state (or state and parameters) at a certain time, taking the underlying model predictive and observational uncertainties into account. In the approach presented in the paper, only model parameter uncertainty is accounted for, and the results clearly show that this is not sufficient to produce skilful predictions from the filtering. These results are not surprising, since the filter does not explicitly account for one of the major error sources in hydrological modelling, the error in the precipitation forcing. Data assimilation methods that update the model state directly account for this error.

Author response: The concern raised by the reviewer points to two sentences of relatively minor significance to the paper. These two sentences will be removed:

The following sentence in section 1 (p2, lines 19 – 21) will be removed. "One drawback of traditional DA (of states only) and of the aforementioned parameter and state DA methods, however, is that the resulting parameters and states are not necessarily compatible with one-another."

The following sentence in section 4.4 (p18, lines 6 and 7) will be removed. "Another advantage is that the parameters and state variables are always consistent with one-another. This cannot be said for other approaches such as the dual Particle Filter or dual Ensemble Kalman Filter."

2. The filtering approach developed uses a 3-day window to select the top 10 best parameter sets. The use of a 3-day window is not justified, and it seems questionable whether such a short window is sufficient considering the different time scales of runoff responses, ranging from slowly varying baseflow to fast responding overland flow contribution. The optimal window size will depend on flow regime.

Author Response: Our response here is the same as for the first reviewer, and is repeated below:

This is an excellent point. The relatively small size of the basin (1,324 $km^2$) provides

some justification for smaller time periods to be used as filters. The 3-day filter would likely be completely unsuitable for basins that are much larger. In addition, the results of the parameter and preceding streamflow filter illustrate that obtaining reasonable results during rainfall events require more than the 3-day filter of streamflow alone. The reviewer's point that parameters that produce good baseflow are rarely the same parameters that produce good flood peaks provides justification for an application of the P-SEDA method that includes parameter sets that produce good baseflow as well as parameter sets that produce good flood peaks, or more likely parameter sets that produce simulations that capture the transition between low and high flows.

The final sentence of the third paragraph in section 4.2 will be changed to include "...
rain during dry conditions, baseflow conditions, flood peaks, transitions between low and high flows, or whatever else..."

In addition, the following paragraph after the third paragraph in section 4.2 will also be added: "The relatively small size of the basin (1,324 $km^2$) provides some justification for the relatively small time period of three days to be used as a filter. The 3-day filter would likely be completely unsuitable for basins that are much larger. However, the results of the parameter and preceding streamflow filter illustrate that obtaining reasonable results during rainfall events requires more than the 3-day filter of preceding streamflow alone. One possible solution to this problem is to have a longer filter period, but other options also exist, as described in the following paragraphs."

Finally, a short paragraph will be added at the end of section 4.2. "All such possible methods to improve the P-SEDA approach require further examination that is beyond the scope of this paper."

3. A latin hypercube sampling approach is applied for generating the population of parameter sets from which the top 10 parameter sets are selected in the filtering approach. The authors discuss the limitation of the LHS approach. I wonder why this limitation has not been addressed in the work. The results of the bulk calibration filter

that corresponds to a classical calibration-validation approach clearly show the limitation of the LHS approach.

Author response: This limitation will be addressed in future work.

Detailed comments 1. Page 2, line 19-21. Not clear exactly what you mean by this statement (see General comments above).

Author response: This sentence is relatively insignificant for the paper and will be removed. Please see response above.

2. Page 5, line 17-18. Explain "CLASS tile" and "GRU".

Author response: This sentence is relatively insignificant for the paper and will be removed.

3. Page 6, line 1-2. How were the parameters and parameter intervals chosen for the LHS sampling? Based on a preliminary sensitivity analysis?

Author response: An earlier version of the paper included more details about the simple study conducted to select the parameters. A small amount of detail will be reinstated by adding the following sentence at the end of the paragraph on page 6, line 2. "The parameters that were perturbed were based on the lead author's experience with the model. Parameter intervals were set based on the ranges found in sources identified under the source column of Table 3. In the case of User specified parameters, these were set by the lead author."

4. Page 6, line 11. Abbreviation "H-EPS" not defined.

Author response: Thank you for catching this. H-EPS on Page 6, line 11 will be replaced with "Hydrological-Ensemble Prediction System (H-EPS)"

5. Page 6, line 19-20. How did you justify that the choice of the 10 best parameter sets is optimal?
[Figure]

Author response: The choice of the 10 best parameters was arbitrary and is deemed to illustrate the P-SEDA methodology The following sentence will be added on page 6, line 20. "(The number 10 was arbitrarily chosen because it illustrates the P-SEDA methodology.)"

6. Page 7, line 9. Abbreviation "CaPA" not defined.

Author response: CaPA is defined on page 5, line 25. No changes to the text will be made.

7. Page 10, line 8-13. A long explanation. Rephrase.

Author response: We will rephrase the entire paragraph as follows:

"Note that these periods do not necessarily correspond to the rising-limb and recession periods of the hydrograph since the river does not always respond strongly to the precipitation for the time period of study in this basin. As a result, for lack of better terminology, these periods are hereafter referred to as "rain-influenced" and "rain-free". We ask for your indulgence in the potentially confusing use of the terms "rain-influenced" and "rain-free." It would be more correct to say "periods during and immediately after the rainfall within the 3-day period" and "otherwise rain-free," but this terminology would be cumbersome throughout the remainder of the paper."

8. Page 10, line 20-25. Include a paragraph where you introduce the test period and test events.

Author response: The existing (short) paragraph will be altered as follows:

"Recall that MESH is run in a continuous simulation mode for the period of June 2002 to November 2014, with a more detailed analysis of the ensemble selection methodologies from June 1 to October 31, 2014. Within this time period, there are five significant precipitation events. The beginning and ending of the precipitation events are considered as follows:"

9. Page 10, line 26-29. Description of the reference forecast is out of place here. Move to the previous section where it is already introduced (page 9, line 5-6).

Author response: The following sentences will be removed from Page 10, line 26-29

"The skill is calculated with an unskilled reference forecast, which in this study is taken to be the measured streamflow at 00 UTC and 12 UTC each day as the forecast for the next 72 hours. This reference forecast is a persistence forecast, which assumes the streamflow is persistent for the forecast period."

and inserted in the first paragraph of page 9 as follows:

"First, a qualitative analysis is undertaken to take advantage of the human brain's ability to synthesize information. The results are then quantitatively verified using the Ensemble Verification System (EVS, Brown et al., 2010). To examine the quality of the ensemble mean when compared with the corresponding observation, the mean error (ME) is calculated. Then the quality of the ensemble distribution is calculated using rank histograms. Finally, the skill relative to using the current streamflow as the forecast is calculated using the mean Continuous Ranked Probability Skill Score (CRPSS). The unskilled reference forecast in this study is taken to be the measured streamflow at 00 UTC and 12 UTC each day as the forecast for the next 72 hours. This reference forecast is a persistence forecast, which assumes the streamflow is persistent for the forecast period."

10. Page 11, line 27-29. Not clear how the water storage value is calculated. Is it a state variable in the model? Or is it assessed using the water balance calculations described in the discussion?

Author response: This storage is a state variable in the model. The text on Page 11, line 28 will be changed by replacing the words "... water storage values for each... " with "water storage state variables for each..."

11. Page 13, line 4-6. Not clear how the 91 parameter sets are chosen. And how can

this approach be applied in an operational setting?

Author response: The 91 parameter sets are chosen by confining the values of the normalized parameters based on the author's interpretation of Figure 6. The text will be adjusted as follows:

"Based on a subjective visual analysis of these box-plots, the 10,000 parameter sets are reduced to 91 parameter sets by confining the values of the normalized parameters as follows..."

This approach cannot be applied in an operational setting and simply provides some assurance that the method has the possibility of being useful, as discussed in section 4.2.

12. Page 13, line 11 and line 21. Use "reference forecast" instead of "unskilled forecast".

Author response: "reference forecast" will be used instead of "unskilled forecast."

13. Page 15, line 28-29. An example of using SMOS for DA in a hydrological model can be found in Ridler et al. (2014).

Author response: Thank you for this reference. It will be added to the list of references in the paper.

14. Page 16, line 8-10. The use of LHS is identified as one of the key limitations of the approach developed. So why wasn't this issue further investigated (see General comments above)?

Author response: Please see response to the General comment above.

15. Page 18, line 6-7. Why is it an advantage that parameters and state variables are consistent (see General comments above)?

Author response: Please see response to the General comment above.

16. Page 18, line 12. Abbreviation "H-LSS" not defined.

Author response: H-LSS is defined on page 2, line 3.

17. Tables 2-3. Very detailed information, and difficult to understand without knowledge of the model applied. I suggest to move this to Supplementary material together with a brief description of the model applied.

Author response: There is a very brief description of the model in section 2.3. Tables 2-3 can be moved to supplementary material if required.

18. Table 5, caption. Delete "low-skill".

Author response: "low-skill" will be deleted in the caption of Table 5.

―――――――――――――――――

---

## Author Comment (AC3) · 18 Feb 2018

hess-2017-482: Authors' responses to Reviewer Comments for "Parameter-state ensemble data assimilation using Approximate Bayesian Computing for short-term hydrological prediction"

Responses to RC3

1. Section 2.1: The first few sentences describe briefly how the P-SEDA filter works. States and parameters are drawn from some multivariate initial distribution - and then analyzed for use in a projection period. How is this analysis done? I think a Figure

may really help to communicate to readers how the P-SEDA method is implemented. The two sentences, "The analysis is completed and the process repeated for the next appropriate time-step in the continuous simulations" and "In this manner, both the parameters and states are drawn from the entire M simulations for the projection period". Not clear to me.

Author response: A figure will be added in section 2.1, which will now open with "The P-SEDA filter works in the following manner, as illustrated in Figure 1." We trust that this new figure, along with the second algorithm added to section 2.7 (as requested by the first reviewer), clarifies how the P-SEDA method is implemented.

2. Line 10: so you are talking here about the normalized weights. A particle filter uses three different weights: incremental weight (for current datum only), unnormalized weight (normalized weight prior to datum x incremental weight -> summarizes weight of entire trajectory) and normalized weights -> normalization of unnormalized weights before moving on to the next datum.

Author response: It is not clear that this comment requires a response or any changes to the paper. We do not plan to make any changes to the text accordingly.

3. Line 12: Resampling is the crux to an efficient implementation of the particle filter. Otherwise, many trajectories will receive a negligible weight and the PF does not approximate closely the target PDF.

Author response: It is not clear that this comment requires a response or any changes to the paper. We do not plan to make any changes to the text accordingly.

4. Line 14 - 16 "The approach presented here is the same, but without resampling and always returning to the original particles as updated by the model and assigning a weight of zero or one to each particle based on the filter (i.e. using a rectangular filter)." is unclear to me. This goes back to my earlier comment. From what is presented, I do not understand how the authors implement such approach. Thus, no resampling is

done? How do you return to the original particles. As with comment 1 above, can you give a detailed example, in text or in Figure that explains how this works. For example, at a time, t, we have the state forecast and associated parameter values + an incoming observation. What does the filter do then? How does it return to the original particles? How are the weights assigned? How is resampling avoided? etc.

Author response: We always return to the original particles by performing a continuous simulation of all of the particles. We trust that the addition of Figure 1 and Algorithm 2 makes the process more clear.

5. Algorithm 1: How is $s(y_i)$ computed? And how do you find the theta's from the $k_m$ nearest neighbors of $s_0$?

Author response: In the pure form of ABC, $s(y_i)$ is simply a statistical property, such as mean or standard deviation, of the simulation. This is compared to the same statistical property of the observation ($s_0$). So if the mean were the statistical property being compared between the simulation and the observation, then the $k_m$ nearest neighbors of $s_0$ would be the $k$ simulations that have the mean that is closest to the observations. As with the response to the first major concern of reviewer one, we trust that the addition of Algorithm2 clarifies what is being done in this paper.

6. I do not understand where ABC comes in. Is this in the selection of $s_0$? and the $s_y$'s? And how is the likelihood function formulated? This is done by simulation, yet, I miss the details necessary to understand and comprehend what has exactly been done.

Author response: All of algorithm 1 represents the ABC algorithm. In addition to the previous response, the likelihood function is approximated by the model. The additional text and figures added to the paper (in response to your concerns 1 and 8) should make it easier to understand and comprehend what has exactly been done.

7. Latin Hypercube sampling is argued as being highly inefficient. That is true if you want to approximate a target PDF, nevertheless, if you just want to sample the parameter space, then this may be one of the best methods you can use.

Author response: As stated in the introduction, operational models are generally concerned with predictive ability and thus are more concerned with approximating a target PDF rather than sampling the parameter space. As such, future work in this area should consider methods other than LHS. No changes are planned to address this comment.

8. In Section 2.7 the authors describe how they construct the ensemble. None of the four approaches listed are described in detail. Hence, I do not understand what is being done. "minimized uncertainty filter". Need a detailed explanation, step by step how we go about initial states and parameters to a minimized uncertainty filter. Same holds for the other three listed methods. Without this the results in this paper will not be understood, nor are impossible to be reconstructed by the reader.

Author response: The following Figures will be added to Section 2.7 as Figures 4 to 7 in the transcript, but as Figures 2 to 5 in this response, which we expect will make the details of the various approaches more clear.

[Figure]

[Figure]

**Fig. 1.** Schematic of the Parameter-State Ensemble Data Assimilation (P-SEDA) filter.

[Figure]

**Fig. 2.** Schematic of the Bulk Calibration Filter in which the top 10 simulations are selected from the filter (calibration) period of Jan 1, 2002 to Dec 31, 2009 and run for the projection (validati

[Figure]

**Fig. 3.** Schematic of the Minimized Uncertainty Filter in which the 3-day filter and projection periods are the same for each window of time.

[Figure]

**Fig. 4.** Schematic of the Preceding Streamflow Filter in which each 3 day filter period is followed by a 3 day projection period.

[Figure]

**Fig. 5.** Schematic of the Parameter and Preceding Streamflow Filter. Note that the time period is re-run for the 3 day filter and projection periods after initially reducing the 10,000 parameter sets to 91.

---

## Author Response (AR1)

**Author Response to Editor Decision: Reconsider after major revisions (further review by editor and referees)** (18 Feb 2018) by Harrie-Jan Hendricks Franssen

Comments to the Author:

Dear Dr Davison,

Your manuscript "Parameter-state ensemble data assimilation using Approximate Bayesian Computing for short-term hydrological prediction" has been subjected now to review by three reviewers. Two of them recommended major revision and one of them minor revision. I think the paper can be published after major revision including additional review. However, given the reviewer comments, rejection of the manuscript is still likely if the concerns of the reviewers are not resolved.

The main points to be handled are:

1. Clarification of the methodology at several points, as indicated by the reviewers (especially reviewer #3). Advantages compared to other methods should be clarified. This means that the discussion of the methodology should be placed in a broader context.

Author response: The methodology has been clarified at several points. Broader context has been provided in the introduction (p2, lines 6 – 13), section 2.1 of the methodology (p3, lines 12 – 25), and throughout section 2.7 explaining the ensemble selection methodologies (p6 – 9, and the additions of Figures 1, 2, 3 and 5).

2. The use of a 3-day windows only to select parameters should be justified. This is typically not enough to characterize both base flow and peak flows. The selected example is therefore unfortunate.

Author response:  The analysis has been expanded to also examine 10, 20, 30 and 40 day windows.

3. A better justification of the selected ten parameters is needed. This could for example be done by providing additional sensitivity analysis.

Author response: The analysis has been expanded to include a selection of 5, 10, 20, 30, 40 and 50 parameters based on the lowest RMSE values.

In your answer to the main points and detailed comments, please indicate how comments have been handled exactly, indicating also whether text has been deleted and what the position of newly included text blocks is. I am looking forward to the new version of the paper.

Author response: The changes are too numerous to include here and a document with changes tracked has also been uploaded.

**hess-2017-482: Authors' responses to Reviewer Comments for "Parameter-state ensemble data assimilation using Approximate Bayesian Computing for short-term hydrological prediction"**

**Responses to RC1**

Thank you for the positive "overall recommendation." In answer to your major and minor concerns:

Major Concerns

1) I had to go back to the methods sections several times to interpret what methods were covered by "P-SEDA" and how the method was applied. Algorithm 1 is extremely helpful, but I would suggest a second algorithm or a flow chart that is specific to the P-SEDA method applied, and perhaps specific to the 3-day moving window application presented in the manuscript. The manuscript should be more explicit about how P-SEDA is different from Algorithm 1.

Author Response: A second algorithm is added in section 2.7 to illustrate how the P-SEDA method is applied and how this is different than Algorithm 1. New Figures (1, 2, 3 and 5) have also been added to illustrate the algorithm and its implementation.

2) I cannot tell from the description in the methods section if the filters are applied sequentially or if they always draw from the same set of 10,000 simulations. This is stated more clearly in the conclusions. Because the majority of particle filters are applied sequentially, it should be clear early on that this is not the case in this paper.

Author Response: Some additional context has been provided in the introduction, section 2.7 and conclusions that describe the approach as a hybrid of the particle filter and variational DA methods.

3) I would not expect three days of streamflow to be enough to determine reasonable streamflow parameters. The parameters that produce good baseflow are rarely the same parameters that produce good flood peaks. Please provide more justification for testing this method

Author Response: This is an excellent point. The analysis has been expanded to include longer filter periods.

**Minor concerns**

1) Tables 4 5, lines 10-25, p. 13, and Figure 8, Conclusions: terminology suddenly changes. "Projection methods" were never defined; previously the four P-SEDA methods were all referred to as "filters". It is unclear which is the "3-day filter" and which are projections. I assume they correspond to the previously defined filters as follows, but I am not certain:
a. 3-day filter = minimized uncertainty filter
b. 3-day projection = preceding streamflow filter
c. bulk projection = bulk calibration filter
d. 3-day projection with constrained parameters = parameter and preceding streamflow filter.

Author Response: The terminology has been changed to explain the ensemble selection methodologies more clearly. The minimized uncertainty filter is now called the "optimal hind-cast," the bulk projection has been removed, and the parameter and preceding streamflow filter has been renamed the "hindsight parameter constraint and preceding 3-day streamflow filter."

2) Line 21, p. 6: Instead of saying "ensemble data assimilation filters", say "P-SEDA" filters. Otherwise, it is not clear that P-SEDA encompasses all 4 filter approaches.

Author Response: The change in text has been made as suggested.

3) Lines 30-31, p. 13: I assume that you do not use the bulk calibration filter here because it performed poorly, but it is probably worth stating that.

Author Response: Yes. This is correct. However, in this latest version of the paper, the bulk calibration filter has been removed as an example.

4) Line 32, p. 17: The reason you chose to focus on 2014 should be stated in methods.

Author Response: The last two paragraphs of section 4.3 will be moved to the end of section 2.8 in the methods.

5) How do the authors propose to implement the parameter and preceding streamflow filter? Operationally, one would not extract parameters for all days during a year's precipitation events before setting the parameter range for this filter. Would it be based on the previous year's filter or would the parameter prior distribution be updated based on the days leading up to the current date? How would that impact the result? This is touched on in the conclusions, and it is probably beyond the scope of this study, but it would be helpful if this limitation were mentioned in the methods section.

Author Response: Yes. The implementation of this filter in an operational setting is not feasible. It was mainly included to show that a small subset of the 10,000 LHS runs could be used more effectively than the full 10,000 parameter sets. The final sentence of section 2.7.3 is our response to this comment: "This ensemble represents an approach that cannot be used in a forecasting context, but does represent a proxy for other parameter-constraining methods that are explored in the discussion."

6) Line 8, p. 19: Shouldn't it be the minimized uncertainty filter that shows the model is capable of simulating streamflow for any 3-day period.

Author Response: Yes. This has been corrected in the text.

**Typos, grammar, etc.**
1) Line 23, p. 9: "eror" -> "error"

Author response: This will be corrected.

2) Line 25, p. 13: I'm pretty sure you mean 4c to 4e and 7.

Author response: This has been corrected.

3) Lines 20-21, p. 15: ": : :including the state of basin storage in the assessment of equifinality clearly shows that the parameter-state sets are not equal." I'm not sure what is meant by "assessment of equifinality." Also, please refer either to a figure, a table or a citation that supports this.

Author response: Thank you for pointing this out. This sentence is not clear and the paragraph will be adjusted as follows:

"The issue of widely-varying simulated basin storage (Figure 4c) also highlights the issue of equifinality, which is defined here as the idea that many different model simulations can produce acceptable results (Beven, 1993). The model is able to find many parameter-state sets that fit the streamflow for short periods of time. If only streamflow observations are available, the selected simulations are equifinal. However, including the state of basin storage clearly shows that the parameter-state sets are not equal. If soil moisture observations are also available and used, then these simulations are not equifinal and the selected simulations can be further constrained."

4) Figure 1: Color of lakes and rivers in legend should match their color in Fig. 1c.

Author Response: The figure has been changed accordingly.

5) Figure 2: The toolbar at the top should be removed.

Author Response: The figure has been changed accordingly.

6) Figure 5: Plot storage on the same scale in a and b. I'm guessing the solid black line is precipitation and the dots are storage simulated by the best 10 parameter sets. A legend would help.

Author response: The figure has been changed accordingly.

7) Table 4: "assessement" -> "assessment"

Author response: This has been corrected.

**hess-2017-482: Authors' responses to Reviewer Comments for "Parameter-state ensemble data assimilation using Approximate Bayesian Computing for short-term hydrological prediction"**

**Responses to RC2**

Major Concerns

1. The authors state that a drawback of traditional filtering methods that update the model state only or update state and parameters simultaneously provide state and parameter values that are not consistent (described in the Introduction, and re-iterated in the Discussion). But this "inconsistency" is the exact nature of filtering-based data assimilation as opposed to variational approaches and smoothing methods. The objective of filtering is to provide the best estimate of the state (or state and parameters) at a certain time, taking the underlying model predictive and observational uncertainties into account. In the approach presented in the paper, only model parameter uncertainty is accounted for, and the results clearly show that this is not sufficient to produce skilful predictions from the filtering. These results are not surprising, since the filter does not explicitly account for one of the major error sources in hydrological modelling, the error in the precipitation forcing. Data assimilation methods that update the model state directly account for this error.

Author response: The concern raised by the reviewer points to two sentences of relatively minor significance to the paper. These two sentences will be removed. Additional context has been provided to describe the approach as a hybrid of particle filter and variational DA. We disagree that the approach only accounts for model parameter uncertainty. The P-SEDA approach also accounts for changes to the model states by selecting joint parameter-state sets that vary both states and parameters for each selected parameter set (see Figure 7c for example).

2. The filtering approach developed uses a 3-day window to select the top 10 best parameter sets. The use of a 3-day window is not justified, and it seems questionable whether such a short window is sufficient considering the different time scales of runoff responses, ranging from slowly varying baseflow to fast responding overland flow contribution. The optimal window size will depend on flow regime.

Author Response: This is an excellent point. The analysis has been expanded to include longer filter periods.

3. A latin hypercube sampling approach is applied for generating the population of parameter sets from which the top 10 parameter sets are selected in the filtering approach. The authors discuss the limitation of the LHS approach. I wonder why this limitation has not been addressed in the work. The results of the bulk calibration filter that corresponds to a classical calibration-validation approach clearly show the limitation of the LHS approach.

Author response: We would have to re-do the entire study from scratch to address the limitation in this study. Given the limitation of LHS, however, it is reassuring that the approach works as well as it does with the longer filter periods. This limitation will be addressed in future work.

Detailed comments

1. Page 2, line 19-21. Not clear exactly what you mean by this statement (see General comments above).

Author response: This sentence is relatively insignificant for the paper and will be removed.

2. Page 5, line 17-18. Explain "CLASS tile" and "GRU".

Author response: This sentence is relatively insignificant for the paper and will be removed.

3. Page 6, line 1-2. How were the parameters and parameter intervals chosen for the LHS sampling? Based on a preliminary sensitivity analysis?

Author response: An earlier version of the paper included more details about the simple study conducted to select the parameters. A small amount of detail will be reinstated by adding the following sentence at the end of the paragraph on page 6, line 2. "The parameters that were perturbed were based on the lead author's experience with the model. Parameter intervals were set based on the ranges found in sources identified under the source column of Table 3. In the case of user specified parameters, these were set by the lead author."

4. Page 6, line 11. Abbreviation "H-EPS" not defined.

Author response: Thank you for catching this. H-EPS on Page 6, line 11 will be replaced with "Hydrological-Ensemble Prediction System (H-EPS)"

5. Page 6, line 19-20. How did you justify that the choice of the 10 best parameter sets is optimal?

Author response: The choice of the 10 best parameters was arbitrary. We have expanded the analysis to include 5, 10, 20, 30, 40 and 50 parameter sets. The method did not show much sensitivity to the number of parameter sets.

6. Page 7, line 9. Abbreviation "CaPA" not defined.

Author response: CaPA is defined on page 5, line 25. No changes to the text will be made.

7. Page 10, line 8-13. A long explanation. Rephrase.

Author response: We will rephrase the entire paragraph as follows:

"Note that these periods do not necessarily correspond to the rising-limb and recession periods of the hydrograph since the river does not always respond strongly to the precipitation for the time period of study in this basin. As a result, for lack of better terminology, these periods are hereafter referred to as "rain-influenced" and "rain-free". It would be more correct to say "periods during and immediately after the rainfall within the 3-day period" and "otherwise rain-free," but this terminology would be

cumbersome throughout the remainder of the paper. Furthermore, it is also important to note that the terms "rain-influenced" and "rain-free" only refer to a time period rather than the discharge of the river. The time periods that these terms refer to are the stretch of time under consideration in the analysis."

8. Page 10, line 20-25. Include a paragraph where you introduce the test period and test events.

Author response: The existing (short) paragraph will be altered as follows:

"Recall that MESH is run in a continuous simulation mode for the period of June 2002 to November 2014, with a more detailed analysis of the ensemble selection methodologies from June 1 to October 31, 2014. Within this time period, there are five significant precipitation events. The beginning and ending of the precipitation events are considered as follows:"

9. Page 10, line 26-29. Description of the reference forecast is out of place here. Move to the previous section where it is already introduced (page 9, line 5-6).

Author response: The following sentences will be removed from Page 10, line 26-29

"The skill is calculated with an unskilled reference forecast, which in this study is taken to be the measured streamflow at 00 UTC and 12 UTC each day as the forecast for the next 72 hours. This reference forecast is a persistence forecast, which assumes the streamflow is persistent for the forecast period."

and inserted in the first paragraph of page 9 as follows:

"First, a qualitative analysis is undertaken to take advantage of the human brain's ability to synthesize information. The results are then quantitatively verified using the Ensemble Verification System (EVS, Brown et al., 2010). To examine the quality of the ensemble mean when compared with the corresponding observation, the mean error (ME) is calculated. Then the quality of the ensemble distribution is calculated using rank histograms. Finally, the skill relative to using the current streamflow as the forecast is calculated using the mean Continuous Ranked Probability Skill Score (CRPSS). The unskilled reference forecast in this study is taken to be the measured streamflow at 00 UTC and 12 UTC each day as the forecast for the next 72 hours. This reference forecast is a persistence forecast, which assumes the streamflow is persistent for the forecast period."

10. Page 11, line 27-29. Not clear how the water storage value is calculated. Is it a state variable in the model? Or is it assessed using the water balance calculations described in the discussion?

Author response: This storage is a state variable in the model. The text on Page 11, line 28 will be changed by replacing the words "… water storage values for each… " with "water storage state variables for each…"

11. Page 13, line 4-6. Not clear how the 91 parameter sets are chosen. And how can this approach be applied in an operational setting?

Author response: The 91 parameter sets are chosen by confining the values of the normalized parameters based on the author's interpretation of Figure 6. The text will be adjusted as follows:

"Based on a subjective visual analysis of these box-plots, the 10,000 parameter sets are reduced to 91 parameter sets by confining the values of the normalized parameters as follows…"

A new Figure (Figure 5) has also been added to illustrate the approach.

This approach cannot be applied in an operational setting and simply provides some assurance that the method has the possibility of being useful, as discussed in section 4.2.

12. Page 13, line 11 and line 21. Use "reference forecast" instead of "unskilled forecast".

Author response: "reference forecast" will be used instead of "unskilled forecast."

13. Page 15, line 28-29. An example of using SMOS for DA in a hydrological model can be found in Ridler et al. (2014).

Author response: Thank you for this reference. It will be added to the list of references in the paper.

14. Page 16, line 8-10. The use of LHS is identified as one of the key limitations of the approach developed. So why wasn't this issue further investigated (see General comments above)?

Author response: Please see response to the General comment above.

15. Page 18, line 6-7. Why is it an advantage that parameters and state variables are consistent (see General comments above)?

Author response: Please see response to the General comment above.

16. Page 18, line 12. Abbreviation "H-LSS" not defined.

Author response: H-LSS is defined on page 2, line 3.

17. Tables 2-3. Very detailed information, and difficult to understand without knowledge of the model applied. I suggest to move this to Supplementary material together with a brief description of the model applied.

Author response: There is a very brief description of the model in section 2.3. Tables 2-3 can be moved to supplementary material if required.

18. Table 5, caption. Delete "low-skill".

Author response: "low-skill" will be deleted in the caption of Table 5.

**hess-2017-482: Authors' responses to Reviewer Comments for "Parameter-state ensemble data assimilation using Approximate Bayesian Computing for short-term hydrological prediction"**

**Responses to RC3**

1. Section 2.1: The first few sentences describe briefly how the P-SEDA filter works. States and parameters are drawn from some multivariate initial distribution - and then analyzed for use in a projection period. How is this analysis done? I think a Figure may really help to communicate to readers how the P-SEDA method is implemented. The two sentences, "The analysis is completed and the process repeated for the next appropriate time-step in the continuous simulations" and "In this manner, both the parameters and states are drawn from the entire M simulations for the projection period". Not clear to me.

Author response: Additional context describing the filter as a hybrid of the particle filter and variational data assimilation, and a figure will be added in section 2.1. A second algorithm will also be added in section 2.7. We trust that the additional context, this new figure, and the second algorithm added to section 2.7 clarifies how the P-SEDA method is implemented.

[Figure]

Figure 1: Schematic of the P-SEDA filter. M simulations are run continuously from a model, of which the filter chooses a number from which to analyze a projection. The process is then repeated for subsequent filter periods, noting that the M simulations run continuously through the previous projection periods even though they are not all selected for the previous projection period analysis.

2. Line 10: so you are talking here about the normalized weights. A particle filter uses three different weights: incremental weight (for current datum only), unnormalized weight (normalized

weight prior to datum x incremental weight -> summarizes weight of entire trajectory) and normalized weights -> normalization of unnormalized weights before moving on to the next datum.

Author response: It is not clear that this comment requires a response. Any additional thoughts that the reviewer can provide to guide the authors would be appreciated.

3. Line 12: Resampling is the crux to an efficient implementation of the particle filter. Otherwise, many trajectories will receive a negligible weight and the PF does not approximate closely the target PDF.

Author response: The description of the method has been re-worked as a hybrid of the particle filter and variational DA. This will hopefully address the reviewer's comment. If not, any additional thoughts would be appreciated.

4. Line 14 - 16 "The approach presented here is the same, but without resampling and always returning to the original particles as updated by the model and assigning a weight of zero or one to each particle based on the filter (i.e. using a rectangular filter)." is unclear to me. This goes back to my earlier comment. From what is presented, I do not understand how the authors implement such approach. Thus, no resampling is done? How do you return to the original particles. As with comment 1 above, can you give a detailed example, in text or in Figure that explains how this works. For example, at a time, t, we have the state forecast and associated parameter values + an incoming observation. What does the filter do then? How does it return to the original particles? How are the weights assigned? How is resampling avoided? etc.

Author response: We always return to the original particles by performing a continuous simulation of all of the particles. We hope that the addition of Figures 1, 2, 3, 5, Algorithm 2, and the description of the approach as a hybrid DA method makes the process more clear.

5. Algorithm 1: How is s(y_i) computed? And how do you find the theta's from the k_m nearest neighbors of s0?

Author response: In the pure form of ABC, s(y_i) is simply a statistical property, such as mean or standard deviation, of the simulation. This is compared to the same statistical property of the observation (s0). So if the mean were the statistical property being compared between the simulation and the observation, then the k_m nearest neighbors of s0 would be the k simulations that have the mean that is closest to the observations. However, in our approach, we replace a comparison of statistical properties with a cost function (Root Mean Squared Error).

6. I do not understand where ABC comes in. Is this in the selection of s0? and the s_y's? And how is the likelihood function formulated? This is done by simulation, yet, I miss the details necessary to understand and comprehend what has exactly been done.

Author response: All of algorithm 1 represents the ABC algorithm. The likelihood function is approximated by the model. The additional text and figures added to the paper should make it easier to understand and comprehend what has exactly been done.

7. Latin Hypercube sampling is argued as being highly inefficient. That is true if you want to approximate a target PDF, nevertheless, if you just want to sample the parameter space, then this may be one of the best methods you can use.

Author response: Operational models are generally concerned with predictive ability and thus are more concerned with approximating a target PDF rather than sampling the parameter space. As such, future work in this area should consider methods other than LHS. No changes are planned to address this comment.

8. In Section 2.7 the authors describe how they construct the ensemble. None of the four approaches listed are described in detail. Hence, I do not understand what is being done. "minimized uncertainty filter". Need a detailed explanation, step by step how we go about initial states and parameters to a minimized uncertainty filter. Same holds for the other three listed methods. Without this the results in this paper will not be understood, nor are impossible to be reconstructed by the reader.

Author response: The following Figures will be added to Section 2.7, which we expect will make the details of the various approaches more clear.

[revised manuscript text omitted]

---

## Author Response (AR2)

**Editor Decision: Reconsider after major revisions (further review by editor and referees)** (08 Jul 2018)
by Harrie-Jan Hendricks Franssen
Comments to the Author:
Dear Dr Davison,

Your manuscript "Parameter-state ensemble data assimilation using Approximate Bayesian Computing for short-term hydrological prediction" has been subjected now to re-review by the three original reviewers. Two of the three reviews found the manuscript to be improved. One reviewer recommends technical corrections only, another reviewer minor revision (but points in a confidential statement to the editor that the authors provide now not enough motivation anymore for this research, and if this is not solved, the paper should be rejected) and a third reviewer major revision. In summary, the main points to be improved are:
1. Provide a convincing motivation for the research.

I have re-introduced the previous point that the method ensures consistent parameters and state variables. I have also highlighted other unique aspects of this work, such as this is the first application of ABC to an H-LSS for hydrological modelling (as opposed to a more traditional rainfall-runoff model), this is the first application of ABC for short (3-to-40-day) time-slices of a hydrograph, and that this is also the first known application of ABC with an H-EPS (forced with a meteorological EPS).

2. Explain better the methodology, handling all point addressed by reviewer #3.

I now have a better understanding of the points that reviewer #3 has raised. Hopefully my responses address his main concerns.

I suggest major revision to handle these issues. The revised version will again be subjected to review. If the issues are not convincingly solved, the paper has to be rejected as the philosophy of the HESS-journal is not to have repeated major revisions.
In your answer to the main points and detailed comments, please indicate how comments have been handled exactly, indicating also whether text has been deleted and what the position of newly included text blocks is. I am looking forward to the new version of the paper.

Best regards,

Harrie-Jan Hendricks Franssen - editor -
* * *
Report #1
Submitted on 07 Jun 2018
Anonymous Referee #2

**Anonymous during peer-review:** Yes No

**Anonymous in acknowledgements of published article:** Yes No

**Recommendation to the Editor**

**1) Scientific Significance**
Does the manuscript represent a substantial contribution to scientific progress within the scope of this journal (substantial new concepts, ideas, methods, or data)?

Excellent Good Fair Poor

**2) Scientific Quality**
Are the scientific approach and applied methods valid? Are the results discussed in an appropriate and balanced way (consideration of related work, including appropriate references)?

Excellent Good Fair Poor

**3) Presentation Quality**
Are the scientific results and conclusions presented in a clear, concise, and well structured way (number and quality of figures/tables, appropriate use of English language)?

Excellent Good Fair Poor

For final publication, the manuscript should be

**accepted as is**

accepted subject to **technical corrections**

accepted subject to **minor revisions**

reconsidered after **major revisions**

    I am willing to review the revised paper.

    I am **not** willing to review the revised paper.

**rejected**

**Suggestions for revision or reasons for rejection (will be published if the paper is accepted for final publication)**

The comments I had to the first version of the manuscript have been adequately addressed in the revised version. The paper has been significantly improved. It is recommended for publication subject to minor changes and technical corrections given below:

1. Numbering of figures. As a general rule, you number the figures according to the order they are referred to in the text.

All figures are now numbered according to the order they are referred to in the latest draft of the paper.

2. Page 8, line 6. Figure 3 instead of Figure 2.

Thank you for catching this. The Figure numbering has been corrected.

3. Page 17, line 27-30. This 'fill and spill' dynamics is an important point that is not just related to the discussion of H-EPS. I suggest moving this part to Section 4.1. for discussion in relation to model error

and definition of uncertainty in the filter.

The text has been moved to section 4.1 (section 5.1 in the latest draft) as suggested.

4. Figure 5. Use k_M instead of 10 as in the other figures.

The figure has been changed as suggested.
* * *
Report #2
Submitted on 13 Jun 2018
Anonymous Referee #1

**Anonymous during peer-review:** Yes No
**Anonymous in acknowledgements of published article:** Yes No

**Recommendation to the Editor**

**1) Scientific Significance**
Does the manuscript represent a substantial contribution to scientific progress within the scope of this journal (substantial new concepts, ideas, methods, or data)?

Excellent Good Fair Poor

**2) Scientific Quality**
Are the scientific approach and applied methods valid? Are the results discussed in an appropriate and balanced way (consideration of related work, including appropriate references)?

Excellent Good Fair Poor

**3) Presentation Quality**
Are the scientific results and conclusions presented in a clear, concise, and well structured way (number and quality of figures/tables, appropriate use of English language)?

Excellent Good Fair Poor

For final publication, the manuscript should be
**accepted as is**
accepted subject to **technical corrections**
accepted subject to **minor revisions**
reconsidered after **major revisions**
    I am willing to review the revised paper.
    I am **not** willing to review the revised paper.
**rejected**

**Suggestions for revision or reasons for rejection (will be published if the paper is accepted for final**

**publication)**

The revised manuscript is clearer than the previous version. Figures 1-5 are really helpful in explaining the methodology. The authors added a sensitivity test for the number of days used in the preceding streamflow filter and also a sensitivity test to the parameters used. These exercises address the major revisions requested in the previous review. My primary concern at this point is that by removing the previously given justification that P-SEDA ensures internally consistent parameter-state time series, the authors removed most of their justification for the methodology. The remaining justification is the development of a simple and flexible method for easier operational implementation; however, the most effective method presented—the preceding streamflow filter with parameter constraints—cannot be readily applied in an operational setting. More motivation/justification is needed for development of P-SEDA. Also, given that the 20-day preceding streamflow filter performs much better than the 3-day, some justification should be given for continuing to focus on the 3-day filter.

Removing the previous given justification that P-SEDA ensures internally consistent parameter-state time series does not change the fact that the approach ensures internally consistent parameter-state time series. The justification that was removed has been re-introduced.

Regarding the continued focus on the 3-day filter, the justification that I had provided in the paper was that these were the "only filters that showed any skill in the rain-influenced periods" and that I am using them for illustrative purposes. This argument remains true when comparing the 20 day filter with the 3-day filter with parameter constraints (see Table 5). Using this "unforecastable" approach assumes that the limitations with the "forecastable" approaches can be resolved through some of the ideas presented in the discussion of the paper. In addition, a more informative analysis of a full H-EPS that does include precipitation (and other forcing) uncertainty in the analysis can be performed when using the 3-day filter with parameter constraints. The resulting lack of skill in the H-EPS results can then be more fully attributed to model structural and forcing errors. The following sentence has been added at the end of the first paragraph of section 3.5 "H-EPS" of the results section: "Although these two ensembles are "unforecastable," performing this analysis provides a more meaningful mechanism to examine model structural and forcing errors."

It is also worth noting that this is, to the authors' best knowledge, the first application of ABC to an H-LSS for hydrological modelling, as opposed to a more traditional rainfall-runoff model. As well, this is the first application of ABC for short (3-to-40-day) time-slices of a hydrograph. Finally, this is also the first known application of ABC with an H-EPS. A new (2nd) paragraph has been added to the conclusions to this effect.

p. 15, lines 26-27: Even if soil moisture is used as a constraint, a variety of parameter sets could produce similar results. It is a useful constraint, but it does not remove equifinality.

Yes. I agree. The following text has been added at the end of the paragraph: "Of course, including soil moisture observations to further constrain the selection of simulations would not remove equifinality. It would simply make it more likely that the model is more accurately predicting both streamflow and soil moisture."

Report #3

Submitted on 02 Jul 2018
Referee #3: Jasper Vrugt, jasper@uci.edu

**Anonymous during peer-review:** Yes No

**Anonymous in acknowledgements of published article:** Yes No

**Recommendation to the Editor**

**1) Scientific Significance**
Does the manuscript represent a substantial contribution to scientific progress within the scope of this journal (substantial new concepts, ideas, methods, or data)?

Excellent Good Fair Poor

**2) Scientific Quality**
Are the scientific approach and applied methods valid? Are the results discussed in an appropriate and balanced way (consideration of related work, including appropriate references)?

Excellent Good Fair Poor

**3) Presentation Quality**
Are the scientific results and conclusions presented in a clear, concise, and well structured way (number and quality of figures/tables, appropriate use of English language)?

Excellent Good Fair Poor

For final publication, the manuscript should be

**accepted as is**

accepted subject to **technical corrections**

accepted subject to **minor revisions**

reconsidered after **major revisions**

   I am willing to review the revised paper.

   I am **not** willing to review the revised paper.

**rejected**

**Suggestions for revision or reasons for rejection (will be published if the paper is accepted for final publication)**

I was asked to look again at a revision of this paper. To be honest, at this point, I am not sure whether I commented on an earlier draft of this paper. I think I did. In short, I looked at the revision and have substantial concerns about the presented work. I did confirm with the Editor that I am reviewing the "right" version of the manuscript as I was surprised seeing relatively few changes to earlier comments on the methodology.

I feel that a brief note about the background for how this paper came about is important.

It's entirely possible that I'm using terminology that means something very specific, while I have implemented an approach that is actually different. The reason that I did not adequately address your earlier comments was due to the fact that I do not have a strong background in traditional data assimilation methods. I simply noticed that if I ran thousands of simulations of MESH (a physicallybased, deterministic model), it was generally very easy to find simulations that fit each part of the hydrograph. This paper is my first attempt to try and formalize this finding… and in doing so, it appeared that the method had something in common with particle filtering and ABC. There are certainly similarities to GLUE and the Limits of Acceptability (LOA) approach. It could be that this approach has another formal name of which I am unaware. To this end, I have approached a number of people with a stronger background in statistics and have been reading to gain a better understanding of the formal mathematics (and all of the variants) of particle filtering and ABC, along with other potential candidates for what I am doing might be called. Pointing me to BaRE, PIMLI, and DYNIA is very helpful in this regard.

1. The authors use the word "filtering" - this suggests a state estimation exercise. In practice, no effort is made to discuss/describe the filtering methodology. Likelihood, incremental likelihood, and normalized likelihood. Which state variables are being included in the analysis The responses of the authors' to previous comments on this matter were not helpful as they did not lead to changes in the paper. What is the model error that is used - the stochastic perturbation of the state/output forecast? What is the measurement error of the discharge data that is being used - all this determines the likelihood, incremental likelihood and normalized likelihood of the particles - and so their time evolution. This is all crucial information to understand the implementation and evaluate the results. As I guess the authors pick at each time from a predefined ensemble those parameter vectors that perform "best" for a short past time period - and use those with state estimation to generate a forecast. I am not sure about my interpretation.

My reading to date leads me to believe that what I am proposing is more aligned with a plain English use of the word "filter," which is to remove unwanted model runs for an analysis of the predictive ability of such an approach. I now better understand the use of the word "filtering" in the traditional use of particle filters and agree that this is not what I am doing in this paper. I am really "screening" candidate model runs based on a comparison of measured and simulated streamflows, and using the "screened-in" model runs to make predictions. I have updated the text to remove the word "filter" and replace it with something more appropriately descriptive, which will hopefully avoid confusing terminology with traditional particle filters.

2. If filtering is not used re: state estimation then the authors should make this clear. In their algorithmic recipe nothing is mentioned about state estimation - nor details given. So, if filtering refers to picking out the "best" parameters at each time step then this should be made clear. If this is true, then the authors are missing other important work on this topic such as the DYNIA approach - and the PIMLI methodology that have elements in common with what the authors are doing in present paper. What is also relevant is the BARE approach of Thiemann et al. (2000) - Bayesian recursive estimation - also many elements in common with present approach if authors did indeed not do state estimation.

Filtering is not used re: state estimation and I trust this is now clear in the refined paper. I have also added additional references to DYNIA, PIMLI and BaRE.

3. The authors implement an ABC methodology. They use the model as likelihood - but no likelihood is presented - OK - if the model output is the likelihood then this does not suffice - the likelihood (simulated discharge) requires perturbation to make the model stochastic rather than deterministic. ABC method cannot work with a deterministic likelihood - otherwise the posterior would converge to a point in the limit of "epsilon" going to zero. So, what is the stochastic perturbation that the authors

are using to compute/define the likelihood? One cannot simply use the present model as likelihood. Instead, assumptions are required about the expected probabilistic properties of the model error - and one needs to sample from this distribution to corrupt the deterministic forecasts - this will then converge to the exact target - if the assumptions about the residual errors are honored by the data. See Vrugt and Beven (2018) for more detailed comments on this matter - and our earlier papers published in 2013, 2014 and 2015 in WRR, HESS, etc..

In my opinion, the argument that a stochastic model is necessary for ABC is a distraction. It is entirely possible that the method does not represent ABC in a pure form, but that does not detract from the idea that there is a deterministic equivalent to ABC that may be useful. I have tried to gain some insight (through reading and conversation with others) into whether-or-not ABC can be used with a deterministic model. I have to admit that I have not come to a satisfactory conclusion in definitively answering this question. As a result, I have changed the title of the paper to indicate that I am implementing a deterministic equivalent to ABC (as you also do in your publications and have indicated as such). I have also added a section to the discussion asking if a stochastic model is really necessary for real-world applications of ABCDE. Although I risk coming across as confrontational in the text, I am genuinely interested in seeing your response and gaining a better understanding of why the approach I am using may be considered invalid… or perhaps what it should be called if not ABCDE.

I think it's also important that my approach is a different implementation of ABC, where the top X are selected rather than using an epsilon. (Now shown as algorithm 1 and 2 in the paper.) I believe this makes a difference. I don't believe that any model will be sufficiently adequate for the posterior parameter distribution "to shrink in the limit of epsilon going to zero and eventually converge to a Dirac delta function" (p 4836, 2013 paper) Therefore the 2$^{nd}$ algorithm in Biau (2013) is perhaps more appropriate to use than the 1$^{st}$ algorithm.

4. The authors use Latin Hyper Cube Sampling to sample 10,000 parameter vectors - and at each time they simple use for their forecasts the best "M" parameter vectors from the past few days. Is this what the authors are doing? This involves no filtering as per state estimation. If state estimation is used on top then I am worried about the initial states of the members that are not used. Lets say that at time 1 one uses members 1-10 to generate a forecast for time 2 - those members were found to produce the best forecasts for time 0. Then state estimation is used for those members 1-10. Then I would expect the initial states at time 2 of these members to be better than those remaining members (11-10000) of the ensemble s their states have not been estimated - so then at time 2 I would expect the first 10 members to do better for time 3 - as their states were estimated - and not those of the other members. So, again, I have simple but important questions about the methodology.

I agree that what I am doing involves no filtering as per state estimation as indicated in my response to your first comment. I am really making use of the fact that, as a deterministic model operator for the likelihood, the parameters and states are tied to one-another.

I cannot evaluate the results without understanding in detail the methodology.
I hope these comments are useful to further enhance the manuscript. It may very well be possible that I completely misread/misunderstand the authors intentions. Even then, it may benefit the authors from addressing some of these comments/concerns as other readers may experience similar issues with the presented material.

Agreed. Your insistence on accuracy in the language I use in the paper is appreciated. I hope that my edits and responses clarify the methodology to your satisfaction, and by extension to the satisfaction of others who may have similar concerns.

Jasper Vrugta
Irvine, July 2, 2018

[revised manuscript text omitted]

---

## Author Response (AR3)

**Major Revision**

**Editor Decision: Publish subject to revisions (further review by editor and referees)** (25 Oct 2018) by Harrie-Jan Hendricks Franssen
Comments to the Author:
Dear Dr Davison,

Your manuscript "Parameter-state ensemble data assimilation using a deterministic equivalent of Approximate Bayesian Computing for short-term hydrological prediction" has been subjected now to re-re-review by two of the original reviewers and we apologize for the delay. One reviewer recommends technical corrections only, another reviewer moderate revision. It is therefore especially important to handle the comments by the reviewer recommending moderate revision. Moderate revision is still needed to clarify the methodology and improving the reference to existing literature.

In your answer to the main points and detailed comments, please indicate how comments have been handled exactly, indicating also whether text has been deleted and what the position of newly included text blocks is. I am looking forward to the new version of the paper.

**Author response: Please see below for responses and the document with tracked changes for our edits.**

Best regards,

Harrie-Jan Hendricks Franssen - editor -
* * *
**Referee Nomination & Report Request started** (16 Aug 2018) by Harrie-Jan Hendricks Franssen
Minimum Number of Reports Required: 2

Referee #3: Vrugt, Jasper  jasper@uci.edu
nominated 22 Aug 2018, accepted 25 Oct 2018, report 25 Oct 2018  **[Report #2]**

Anonymous Referee #1
nominated 16 Aug 2018, accepted 20 Aug 2018, report 21 Sep 2018  **[Report #1]**

Nominated Referee
nominated 16 Aug 2018, declined 18 Aug 2018

**Uploaded Files validated** (16 Aug 2018) by Lorena Grabowski

**File Upload** (13 Aug 2018) by Bruce Davison    Abstract    Manuscript    Author's Response

**Report #1**

Submitted on 21 Sep 2018
Anonymous Referee #1

**Anonymous during peer-review: Yes** No

**Anonymous in acknowledgements of published article: Yes** No

**Recommendation to the Editor**

| | |
|---|---|
| **1) Scientific Significance**
Does the manuscript represent a substantial contribution to scientific progress within the scope of this journal (substantial new concepts, ideas, methods, or data)? | Excellent **Good** Fair Poor |
| **2) Scientific Quality**
Are the scientific approach and applied methods valid? Are the results discussed in an appropriate and balanced way (consideration of related work, including appropriate references)? | Excellent **Good** Fair Poor |
| **3) Presentation Quality**
Are the scientific results and conclusions presented in a clear, concise, and well structured way (number and quality of figures/tables, appropriate use of English language)? | Excellent **Good** Fair Poor |

For final publication, the manuscript should be

**accepted as is**

**accepted subject to technical corrections**

accepted subject to **minor revisions**

reconsidered after **major revisions**

    I am willing to review the revised paper.

    I am **not** willing to review the revised paper.

**rejected**

**Suggestions for revision or reasons for rejection (will be published if the paper is accepted for final publication)**

The concerns about the justification for this work that I had have been adequately addressed. I am not sure that the updated methodology fully addresses the concerns of reviewer #3, but his review for this portion of the paper.
Line 23 p. 17 (from author response version of manuscript): "If soil moisture observations are also available and used, then the simulations shown in this study would not be equifinal and the selected simulations can could be further constrained. Of course, including soil moisture observations to further constrain the selection of simulations would not remove equifinality." Remove "the simulations shown in this study would not be equifinal and" because it is in direct conflict with the following sentences.

**Author response: The change has been made as suggested.**

**Report #2**

Submitted on 25 Oct 2018
Referee #3: Jasper Vrugt, jasper@uci.edu

**Anonymous during peer-review:** Yes **No**

**Anonymous in acknowledgements of published article:** Yes **No**

**Recommendation to the Editor**

| | |
|---|---|
| **1) Scientific Significance** Does the manuscript represent a substantial contribution to scientific progress within the scope of this journal (substantial new concepts, ideas, methods, or data)? | Excellent Good **Fair** Poor |
| **2) Scientific Quality** Are the scientific approach and applied methods valid? Are the results discussed in an appropriate and balanced way (consideration of related work, including appropriate references)? | Excellent **Good** Fair Poor |
| **3) Presentation Quality** Are the scientific results and conclusions presented in a clear, concise, and well structured way (number and quality of figures/tables, appropriate use of English language)? | Excellent **Good** Fair Poor |

For final publication, the manuscript should be

**accepted as is**

accepted subject to **technical corrections**

**accepted subject to minor revisions**

reconsidered after **major revisions**

    I am willing to review the revised paper.

    I am **not** willing to review the revised paper.

**rejected**

**Suggestions for revision or reasons for rejection** **(will be published if the paper is accepted for final publication)**

1. The authors use the word "Data Assimilation" in the name of their methodology - and also use this term at various places throughout the paper. Yet, their method does not use data assimilation at all. Instead, they now clarified what I thought last time and that is that for the prediction period they simple use the initial states which are linked intrinsically to the parameters. They use a screening period to determine the best N parameter,state combinations to be used for the prediction period. So, in essence this is a down-sampling exercise - those initial state & parameter combinations that are not used in prediction are still evaluated for the same period to get values of the initial states of this realization for the next prediction period (after the current one). In any case, at no point do the authors actually assimilate data - they simply do open loop simulation with M parameter vectors and for prediction they each time pick the N most desirable parameter vectors - derived from an earlier screening period.

   The authors did respond to earlier comments on the use of DA in the title of their methodological framework. I thought I repeat my concerns as the paper is in much better

shape now. Point is - no data is assimilated - the state¶meters are tight together. As this has important ramifications for the presented methodology and introduction this issues needs to be clarified.

**Author response: The words "Data Assimilation" have been removed from the name of the methodology and the text has been changed throughout the paper. The approach is renamed "Parameter-State Ensemble Thinning (P-SET)."**

2. It is still unclear to me after one read why the ABC DE methodology is needed at all. The authors mention ABC - add the deterministic equivalent ( = ABC without stochastic operator) but I am not sure whether this approach is implemented at all. In fact, the authors refer to it on Page 5, Line 3 as a way to interpret their P-SEDA approach. Thus, I am wondering whether the exercise on ABC - with DE in this paper is needed after all. I do not think that Section 2 is needed to provide justification for their P-SEDA method - only if the ABC method is used to find an initial distribution of suitable parameter values that honor some user-defined summary statistics for the entire period.

**Author response: Section 2 has been removed and a cursory discussion of ABC is all that remains. We refrain from calling the method ABCDE.**

3. More fundamental question is whether you can have ABC with a deterministic model operator. In fact, you cannot. The ABC methodology requires use of a stochastic model operator, otherwise it is impossible to approximate the likelihood via repeated simulation.

**Author response: We agree and maintain a discussion of the existing literature related to ABC.**

4. The ABC-DE acronym coined by the authors has been used before by Turner et al. in a publication in Journal of Mathematical Psychology. Here is the link:
http://faculty.sites.uci.edu/bmturner/files/2011/01/Turner.Sederberg.2012.pdf
The ABCDE approach of Turner uses Differential Evolution to sample the ABC posterior; this approach, however, does not work well with increasing number of summary statistics and increasing dimensionality ( = number of parameters) of the target distribution. The DREAM_ABC method of Sadegh and Vrugt provides a remedy against this. The latter is not relevant for this paper - but as paper is cited anyway, I thought I'd mention it here.

**Author response: We no longer refer to the method as ABCDE.**

5. The authors have to fix some literature references - that is - use round brackets if reference appears at end of sentence - and use round brackets around year if authors of paper are part of the sentence. For example, see Line 8 on Page 4.

**Author Response: We believe that we have fixed all of the references.**

Altogether, I would recommend a moderate and final revision to give the authors the opportunity to respond to these remaining and somewhat lingering issues.

[revised manuscript text omitted]